# SlugAtlas, a histological and 3D online resource of the land slugs *Deroceras laeve* and *Ambigolimax valentianus*

Carlos Lozano-Flores[1], Jessica Trujillo-Barrientos[1], Diego A. Brito-Domínguez[1], Elisa Téllez-Chávez[1], Rocío Cortés-Encarnación[1], Lizbeth D. Medina-Durazno[1], Sergio Cornelio-Martínez[1], Alejandro de León-Cuevas[2], Alejandro Ávalos-Fernández[2], Wilbert Gutiérrez-Sarmiento[1], Aldo Torres-Barrera[1], Fernando Javier Soto-Barragán[1], Gabriel Herrera-Oropeza[1¤], Ramón Martínez-Olvera[1], David Martínez-Acevedo[1], Luis C. Cruz-Blake[1], Vanessa Rangel-García[1], Gema Martínez-Cabrera[1], Jorge Larriva-Sahd[1†], Reinher Pimentel-Domínguez[3], Remy Ávila[3], Alfredo Varela-Echavarría[1,2]*

**1** Department of Developmental Neurobiology and Neurophysiology, Instituto de Neurobiología, Universidad Nacional Autónoma de México (UNAM), Querétaro, México, **2** Laboratorio Nacional de Visualización Científica Avanzada (LAVIS), Querétaro, México, **3** Centro de Física Aplicada y Tecnología Avanzada (UNAM), Querétaro, México

† Deceased.
¤ Current address: Centre for Developmental Neurobiology, Institute of Psychiatry, Psychology and Neuroscience, King's College London, London, United Kingdom
* avarela@unam.mx

**Data Availability Statement:** Original sequence data from this work has been deposited in Genbank (PP854456, PP854454, PP854455). All images of

## Abstract

Due to their distinctive anatomical characteristics, land slugs are appealing research subjects from a variety of angles, including stem cell biology, regeneration, embryonic development, allometry, and neurophysiology. Here we present SlugAtlas, an anatomical and histological online resource of the land slugs *Deroceras laeve* and *Ambigolimax valentianus*. The atlas is composed of series of histological sections on the horizontal, sagittal, and transversal planes for both species with 3D viewing tools of their major organs. The atlas was used in this work for a comparative analysis of the major organs and tissues of these two species. We provide a comprehensive histological description of them and an explanation of novel findings of unique features of their anatomy. For *D. laeve*, we additionally studied its ability for degrowth and regrowth, a feature that characterizes animals with high regeneration potential and obtained initial results on the study of the regeneration of its tail. SlugAtlas is a resource that is also built to accommodate future growth and, along with the experimental techniques that we have developed, will provide the foundation for research projects in a variety of biological domains.

## Introduction

The Order Stylommatophora is the most abundant group of land slugs and snails. It belongs to the Subclass Heterobranchia also encompassing marine slugs and snails [1]. Although

the SlugAtlas (https://slugatlas.lavis.unam.mx) can be downloaded from the web page.

**Funding:** This work received funds form Dirección General del Personal Académico (DGAPA-UNAM, https://dgapa.unam.mx) (IN211322 awarded to A. V.-E. and IT101423 awarded to R.A.) and from Consejo Nacional de Humanidades Ciencias y Tecnologías (CONAHCYT, https://conahcyt.mx) (CBF2023-2024-834 awarded to A.V.-E. and FORDECYT-PRONACES/1561826 awarded to R. A.). The following authors received fellowships or support from CONAHCYT: C.L.-F. (PhD), D.A.B.-D. (MSc), J.T.-B. (Ayudante SNI, PhD), S.C.-M. (Posdoctoral), W.G.-S. (Posdoctoral), and R.P.-D. (Posdoctoral). The funders had no role in study design, data collection and analysis, decision to publish, or preparation of the manuscript.

**Competing interests:** The authors have declared that no competing interests exist.

stylommatophorans are adapted to terrestrial life they still require humid environments to reproduce and thrive. This is especially true of slugs which lack the protection of a shell. Slugs of this group are distributed throughout the world and a few European species are now present in diverse world regions, some as pests of economically relevant crops.

Distinctive and synapomorphic anatomical features of stylommatophorans are the suprapedal gland and two pairs of retractile tentacles [2]. The gland is a tongue-shaped structure located in the anterior end of the foot, under the visceral cavity but isolated from it by a membrane. The anterior pair of tentacles, the rhinophores, are olfactory organs, and the posterior pair bear an olfactory organ and the eye and are thus called ommatophores. The lack of an external or prominent shell in land slugs makes them more suitable for diverse laboratory experiments than snails. Some of their anatomical features, however, are related to their evolutionary loss of the shell (limacization). For example, their gonopore is on the anterior right side, the digestive system reaches their caudal end but the anus and excretory pore are located on their right side, approximately at the mid-region adjacent to the air exchange pore, the pneumostome.

Slugs may be used in diverse studies addressing their embryonic development, their unique anatomical features, growth, nutrition, and regeneration. Regarding regeneration, however, most of our knowledge comes from a few species of planaria, hydra, and, among vertebrates, newts and salamanders [3]. Another expression of the high regenerative potential of planarias is their ability to reduce their size upon starvation (degrowth) and to grow again (regrowth) following restoration of feeding [4–6].

Across gastropod mollusks, diverse structures are known to regenerate including the mantle, the foot, the body wall, parts of the cerebral ganglia, and tentacles [3, 7–10]. A marine slug species is even capable of regenerating the whole body from its cephalic region presumably owing to its ability to survive on the energy generated by photosynthetic plastids acquired through cleptoplasty, that is, the incorporation of functional chloroplasts from ingested algae in the gut of the animal [11]. Few studies, however, have addressed the cellular and molecular processes of regeneration in this animal group.

Two species suitable as gastropod study models are the marshland slug *Deroceras laeve* and the three-band garden slug *Ambigolimax valentianus*. The former was originally distributed throughout North America and Northern Europe and the latter is from the Iberian Peninsula [12]. Both species, however, are now found worldwide but a complete histological reference for them is hitherto lacking. Recent studies, in contrast, have described histological atlases for mussels and for the gastropod queen conch [13, 14].

Other gastropod species have been successfully used in diverse research areas. The sea slug *Aplysia californica*, for example, has been a valuable tool in physiological and cellular studies for several decades. Recently, transcriptomic studies of the nervous system and embryological works have also been performed with this species [15–18]. Moreover, the immune systems of the freshwater snail *Biomphalaria glabrata* and the apple snail *Pomacea canaliculata*, have been extensively studied owing to their medical importance as intermediate hosts of human parasites [19–22]. The latter has also been employed in regeneration studies [23, 24].

In this work, we describe aspects of growth and maintenance of *D. laeve* and *A. valentianus* and an anatomical atlas of whole adult individuals of both species in three planes of section, the web-based SlugAtlas (https://slugatlas.lavis.unam.mx). Digitization of histological sections and manual segmentation also allowed the generation of 3D interactive models of their major organs and degrowth and regrowth of *D. laeve* was also observed. Moreover, in an initial application, SlugAtlas was used as reference to analyze the morphological features of the two species and their differences allowing the identification of novel aspects of limacoid slug anatomy. The SlugAtlas was also used to analyze the morphological changes that follow amputation of the

tail of *D. laeve*. Hence, this resource will potentially be useful in various areas of research in gastropods, including anatomy, stem cell biology, regeneration, embryology, and allometry, the control of body proportions.

## Materials and methods

### Animals

Three adult individuals of *D. laeve* and five of *A. valentianus* were collected from urban gardens in Querétaro, Mexico (20˚38'15"N,100˚28'31"W and 20˚41'48"N,100˚28'05"W, respectively) and used to start a colony of each. The technical specifications for use of laboratory animals of the Mexican government (Especificaciones técnicas para la producción, cuidado y uso de los animales de laboratorio, Norma Oficial Mexicana NOM-062-ZOO-1999) and the Guide for the Care and Use of Laboratory of National Institutes of Health in USA do not cover research with these species and no Research Ethics Committee approval is required.

Animals were kept in water-saturated vermiculite in food plastic containers with multiple needle punctures on their lid to allow air exchange and were fed finely ground Laboratory Rodent Diet 5001 (LabDiet). Animal containers were incubated at 18˚C or up to 25˚C with a 12 hour-12 hour light-dark cycle. Prior to fixation, animals were anesthetized for 10–15 minutes submerged in 5% ethanol in water or in a 1:400 dilution in water of a saturated solution of menthol. The saturated solution of menthol was made by adding menthol crystals to absolute ethanol such that an insoluble precipitate remained after vigorous mixing.

### Amputation and F-ara-EdU labeling

Animals were anesthetized for 10–15 minutes in 5% ethanol in water for amputation with a clean cut of a razor blade, after which they were transferred back to normal animal containers. For labeling of DNA synthesis, 5 μl of F-ara-EdU (T511293, Sigma-Aldrich, 1mM in 0.08% DMSO) were injected with a glass pipet in the ventral region of adult animals (5mg of F-ara-EdU were dissolved first in 15 μl of DMSO, then diluted with water to 1mM). After 24 hours, animals were anesthetized, fixed in 4% paraformaldehyde in PBS (Roche, 11666789001) supplemented with 11 mM NaCl, cryoprotected with 30% sucrose in PBS and frozen in blocks of tissue freezing medium (Leica Inc.). Cryosections of 30μm were obtained in a Leica CM3050 Cryostat and mounted on SuperFrost Plus slides (Thermofisher Scientific, www.thermofisher. com). Click chemistry was used to develop F-ara-EdU with Sulfo-Cy5-azide (B3330, Lumiprobe Corporation).

### Histology and electron microscopy

Animals of both species of approximately 200 mg and 500 mg were anesthetized and then fixed in paraformaldehyde. Standard procedures were employed to embed fixed animals in paraffin wax, section (5 μm), and stain with hematoxylin-eosin or Kluver-Barrera. Histological series of whole animals on the transversal, horizontal or sagittal planes stained with hematoxylin-eosin were obtained.

For semithin sections, small fragments of tissue (approximately 3mm on their longest dimension) were fixed in 4% paraformaldehyde and 2% glutaraldehyde in 0.11 M phosphate buffer (pH 7.4) at 4˚C for 2 hours. After 3 washes for 10 minutes each in phosphate buffer, tissue was placed in 1% osmium tetroxide in phosphate buffer at room temperature in gentle rotation for one hour. Osmium was removed with three washes with phosphate buffer for 10 minutes each storing them for about 12 hours at 4˚C, if necessary. Tissue was dehydrated by serial incubation to equilibrium in acetone at 60, 70, 80, and 90%, each made in phosphate

buffer followed by incubation in three changes of absolute acetone for 10 minutes each. Tissues were then incubated in a 1:1 mix of acetone and epon (EMBed 812, Electron Microscopy Sciences, 14120) for 5–16 hours, followed by a 2:1 mix of the same reagents for 5–16 hours. Acetone was evaporated by rotation of the vials containing the tissues without their caps for 3 hours. Tissues were then placed and oriented on electron microscopy molds filled with epon and incubated at 40°C for 16 hours and at 60°C for 24 hours for polymerization of epon. Semi-thin sections (400 nm) were obtained with glass knives in the Ultracut R ultramicrotome (Leica) and stained with toluidine blue.

For transmission electron micrography, ultrathin sections (80–90 nm) were obtained with diamond knives from epon blocks and mounted on 200-mesh copper grids that were previously coated with Formvar film. Sections were contrasted with aqueous solutions of uranium acetate and lead citrate and studied in a JEOL 1010 electron microscope operated at 80 kV and equipped with a Gatan digital camera.

For Scanning Electron Microscopy (SEM) the radulas of both slug species were extracted by dissolving the cephalic region of adult animals in 2% SDS in PBS containing 0.4 mg/ml of Proteinase K for 16–20 hours at 55°C with slow rotation. The radulas were washed with water, mounted on copper sample holders with double sided adhesive tape, vacuum dried, and coated with gold for analysis in a Jeol JSM-6060LV scanning electron microsocope at 20kV (Laboratorio Nacional de Caracterización de Materiales, LaNCaM-CONAHCYT).

## Image collection, processing, and construction of online resources

For the construction of SlugAtlas, each histological section of a series was digitized in multiple micrographs with a 10X objective in a Leica ICC50 HD microscope. A mosaic was assembled manually with the micrographs for each section in Gimp (https://www.gimp.org) and all sections of a series were collated in layers and aligned in register manually also in Gimp. Representative sections of each series were included in the atlas and in Figs 3 and 4.

Micrographs of Figs 6–7, 9–16, and S2 Fig were taken on an Olympus BX50 microscope using a Canon EOS Rebel T3i camera with an in-home 3D-printed microscope adapter. Adjustments of brightness, contrast, color balance, and sharpness were applied to whole micrographs using Gimp and no individual morphological feature was processed independently. Hence, image adjustment did not eliminate or enhance any individual feature in any micrograph.

For the construction of the 3D models, the major organs were manually segmented (ground truth segmentation) on each of the digital sections of the horizontal series for each species (266 sections for *D. laeve* and 216 sections for *A. valentianus*) and 19 stacks of binarized images were obtained, one for each organ (S1A Fig). A 3D gaussian blurring processing with sigma value = 2 was carried out on each structure using ImageJ Fiji (https://imagej.net/software/fiji) (S1B Fig), establishing the continuity between adjacent images of the stack and thus allowing the generation of a smoother surface for each structure upon 3D rendering (S1C Fig). The 3D models were uploaded to a website developed with JavaScript and the rendering library for scientific web visualization was made with vtk.js by Kitware (https://kitware.github.io/vtk-js; https://www.kitware.com) which allows viewing the models from a web browser navigating from the SlugAtlas page (see below) (S1 and S2 Videos). The 3D models of both species are missing some sections of the dorsal aspect of the animals but all organs can nevertheless be visualized.

The SlugAtlas web page (https://slugatlas.lavis.unam.mx) is hosted at the National Laboratory for Advanced Scientific Visualization (Laboratorio Nacional de Visualización Científica Avanzada, LAVIS-CONAHCYT) (https://lavis.unam.mx) and was built using Wordpress/Divi

as the Content Management System (CMS) for management and styling. The atlas visualization application is an in-house web software designed at LAVIS coded with vainilla JavaScript and HTML5, using third party resources only for mobile gestures (hammerjs), for image capturing on site (html2canvas), and for font styling (Google Fonts).

## Cloning and sequencing of Cox1

A fragment of the coding region of Cox1 was amplified by PCR from genomic DNA of two individuals of *D. laeve* and one of *A. valentianus* with the following primers: LCO1490 (5'-GGTCAACAAATCATAAAGATATTGG) and HCO2198 (5'-TAAACTTCAGGGTGACCAA AAAATCA) [25]. PCR fragments were cloned in the pGEM-T Easy vector (Promega Inc., www.promega.com) and sequenced by capillary electrophoresis in an Applied Biosystems AB3730 DNA Analyzer (LANGEBIO, Irapuato, Mexico).

## Multiple alignment of Cox1

The sequences of the cloned Cox1 cDNA fragments of *D. laeve* and *A. valentianus* from this work and other sequences from the same species and from twelve other species of gastropods (S1 Table) were aligned with the BioNJ algorithm for unrooted neighbor-joining tree construction using Seaview [26]. Visualization and refining of the tree was accomplished using iTOL [27].

## Extraction of concretions and analysis by Energy dispersed X-ray spectroscopy (EDS)

The body wall of three *D. leave* adults of approximately 400 mg was dissected out eliminating the cephalic structures and the visceral mass. The tissue was chopped up finely with a razor blade and resuspended in 10 ml of 50 mM Tris-Cl (pH 8.5) containing 0.5% SDS and 70 µg/ml of Proteinase K followed by incubation at 55°C with slow rotation for 16 hours. DNase I was added to a concentration of 100 µg/ml, incubated for 1 hour at 37°C followed by a two-hour incubation at 80°C. Concretions were collected by centrifugation at 4000 xg for 15 minutes, and the pellet was washed three times by re-suspending in 10 ml of water and centrifugation as before. The final pellet was resuspended in 100 µl of water.

Kidneys from two adult individuals were dissected out and the concretions were extracted as described for the body wall but in a volume of 3 ml.

For Energy dispersed X-ray spectroscopy (EDS), droplets containing the concretions were placed on aluminum holders, air-dried, vacuum-dried, coated with gold, and analyzed with a Hitachi SU8230 scanning electron microscope with an EDS Bruker detector (1KV).

## Fluorescence microscopy

Four-month-old specimens were anesthetized, fixed with paraformaldehyde, and processed to obtain frozen sections as described above. Sections were stained with Alexa Fluor 488 phalloidin (1:800 in PBS)(A12379, Invitrogen) and contrasted with 2.5 µg/ml of Hoechst 33342 (14533, Sigma-Aldrich) or SYTOX green (S7020, Invitrogen) diluted 1:2000 in PBS. Confocal images were obtained in an 880 LSM Zeiss microscope with ZEN Black software and captured with Plan-Apo 63x N.A = 1.4 oil immersion objective. For 3D volume reconstruction from confocal Z-stacks (resolution of 1024 x 1024 pixels), imageJ/Fiji software was used followed by Amira 6.4.0 software (Thermofisher Scientific) using the isosurface display function for each fluorescent channel.

## Results

### Laboratory colonies of *Deroceras laeve* and *Ambigolimax valentianus*

As a first step to develop garden slug species as study models, we established colonies of *D. laeve* and *A. valentianus* (Fig 1A and 1B). To confirm the identity of the species in our colony, in addition to the external macroscopic features, we analyzed by scanning electron microscopy the radula, the chitinous rasping structure located in the buccal complex (Fig 1C–1N). We analyzed at various magnifications the denticles of the middle part and edges of the radula, as well as its underside.

Lateral denticles are tricuspid in both species (Fig 1C and 1I). In *D. laeve* the first lateral teeth are tricuspid, with a small endocone close to the tip of the mesocone and an ectocone further away from it. Lateral denticles consist of a single mesocone lobe or have a pointed cusp (Fig 1E–1G). The first lateral teeth in *A. valentianus* are similarly tricuspid, with a prominent mesocone flanked by small and well defined endocone and ectocone, all with tips sharper than in *D. laeve*. Denticles become longer towards the margin of the radular band until they have a single mesocone with a pointed cusp and a serrated inner edge (Fig 1K–1M). Both types of denticles had the morphology described previously for each of these species [12].

Additionally, multiple alignment of a cytochrome oxidase 1 (Cox1) cDNA fragment of both species with other individuals of the same species, along with those of twelve other species of stylommatophorans (S1 Table), revealed that the sequences of our specimens cluster with multiple sequences of *D. laeve* or *A. valentianus* as expected (Fig 1O). More specifically, our *D. laeve* sequences cluster with those of specimens collected in Mexico in a previous study [28].

These species adapted readily to laboratory conditions with no complex husbandry requirements. At 18°C, these hermaphroditic species that require crossed fertilization, have gestation times of approximately 14 days for *D. laeve* and 19 days for *A. valentianus*, reach reproductive age at approximately 3, and 4 months, have average clutch sizes of 36 and 47, and hatching rates of 53% and 35%, respectively (Fig 2B, 2D and 2F).

The growth of *D. laeve* is approximately linear with a minor inflection point at about four months (Fig 2A) reaching a weight of 0.5 g when the animal reaches one year. In contrast, the growth of *A. valentianus* has an initial exponential phase that reaches an inflection point at about three months (Fig 2C). The time of 50% survival in *D. laeve* is 173 days and 154 days for *A. valentiaus* (Fig 2B and 2D). From then on, the growth is linear up to at least one year of age in which it averages 1.5 grams. The growth of both species continues beyond the stage at which it was recorded in our study.

As the ability to degrow and regrow on any species is indicative of a high regeneration potential, we assessed this response for *D. laeve*. Upon starvation, adults of this species reduced their body weight to approximately 35% in 8 weeks at which point they were indistinguishable from normal juveniles of under two months of age (Fig 2E). When these animals were again fed *ad libitum*, an accelerated rate of growth or "rebound" was observed such that the original weight was reached in three weeks.

### SlugAtlas

Whole specimens of *D. laeve* and *A. valentianus* were sectioned in paraffin wax on the horizontal, transversal, or sagittal planes and stained with hematoxylin-eosin (Figs 3A–3C and 4A–4C). The sections of each series were digitized, assembled in mosaic and layered onto a digital image stack (Figs 3D and 4D). As a resource for studies with these species we built the web-based SlugAtlas using representative images of the whole body series of sections on the three planes of section described (https://slugatlas.lavis.unam.mx) (Fig 5A–5D). The web

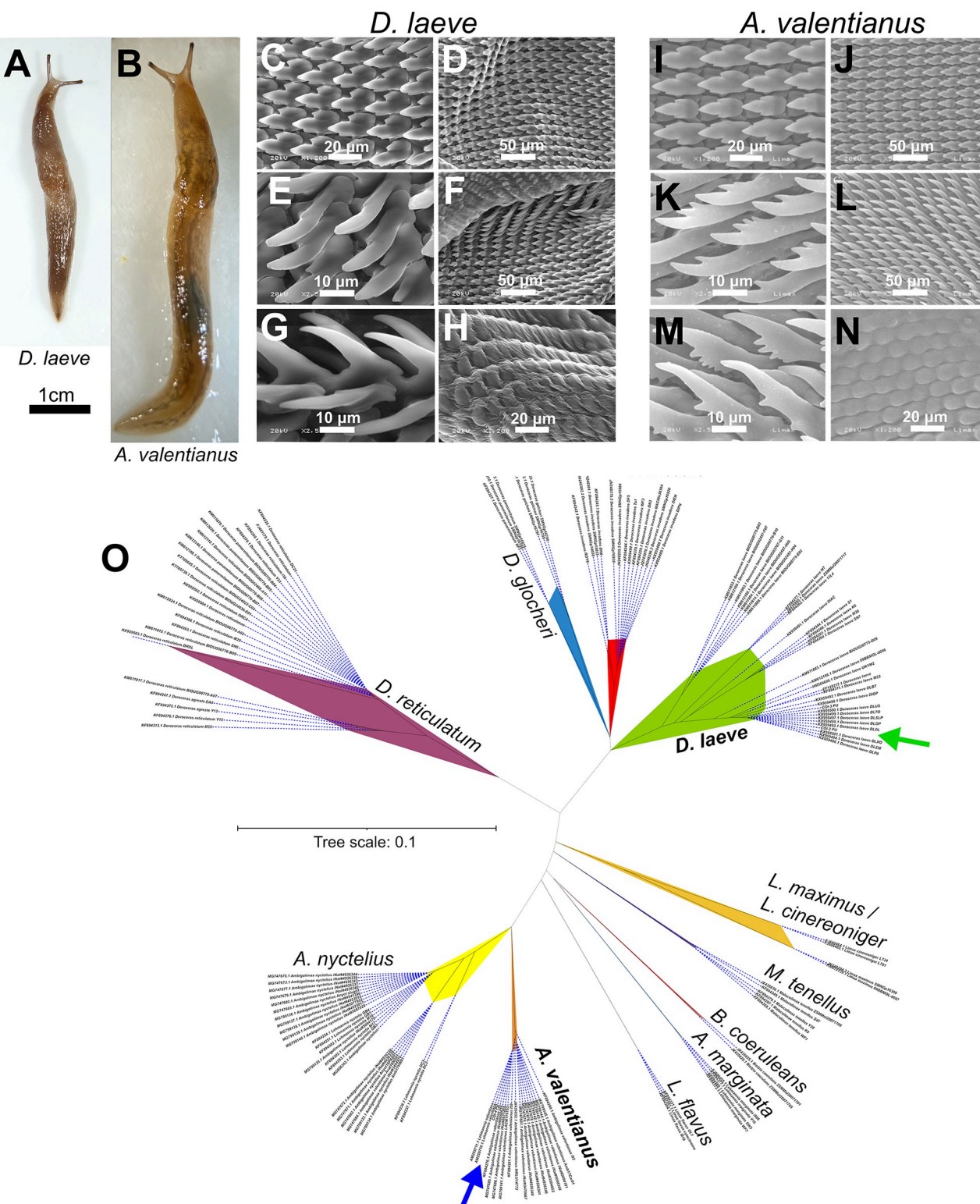

**Fig 1. Classification of *Deroceras leave* and *Ambigolimax valentianus*.** (A) *D. laeve* and (B) *A. valentianus* adult individuals. The radula of *D. laeve* (C-H) and *A. valentianus* (I-N) was analyzed by scanning electron microscopy. (O) The sequence of a fragment of cytochrome oxidase 1 (Cox1) was analyzed by multiple alignment with the corresponding sequences of individuals of fourteen stylommatophoran species including *D. laeve* and *A. valentianus*. Green and blue arrows indicate lab-reared *D. laeve* and *A. valentianus*, respectively, from this work.

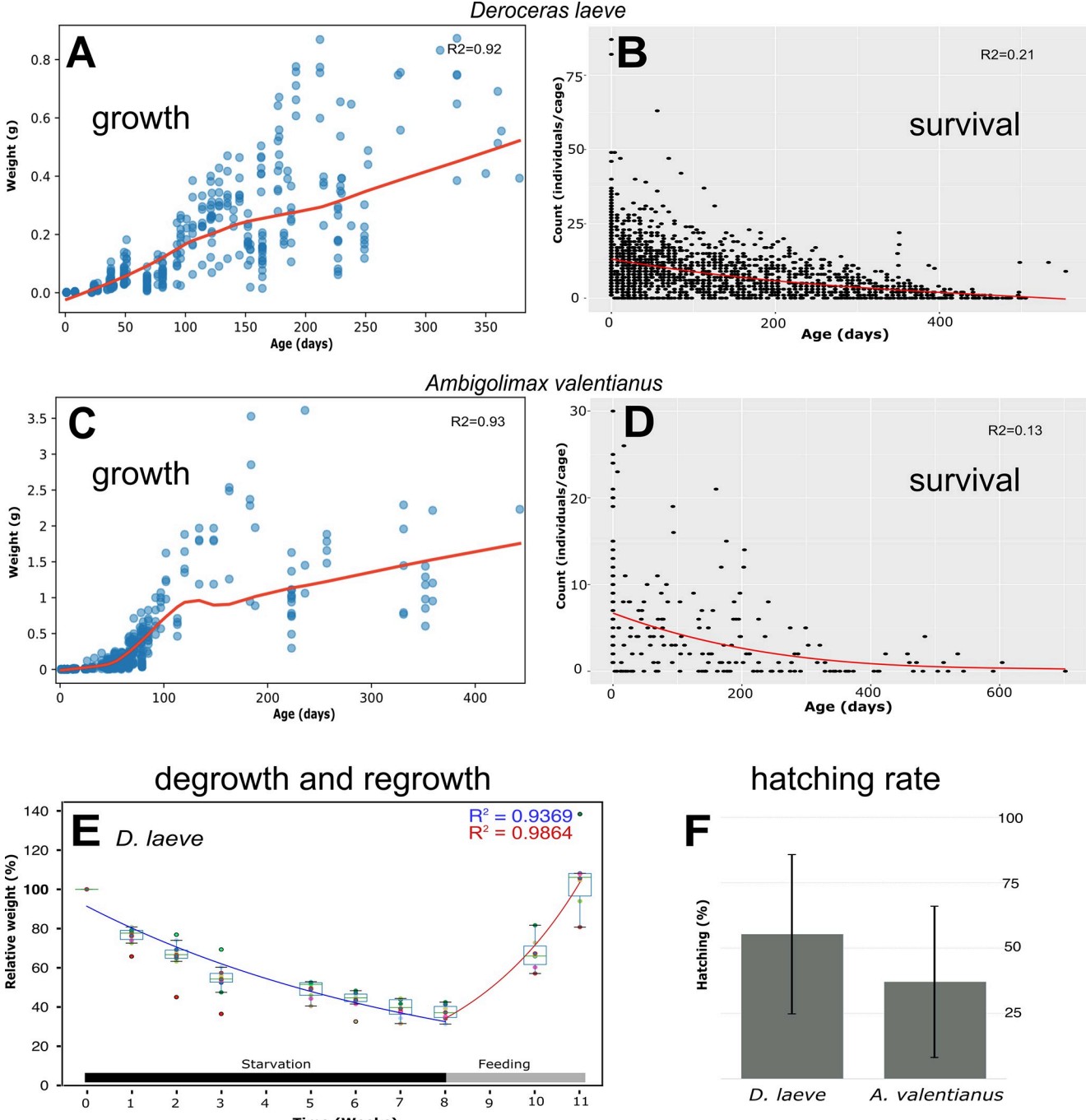

**Fig 2. Growth and reproduction of *D. laeve* and *A. valentianus*.** Growth was measured by weighing individuals each week (A, C), survival rate was determined by counting the members of isolated clutches over time. (B, D), degrowth of *D. laeve* was determined by starvation of animals for eight weeks and feeding *ad libitum* for the subsequent four weeks (E), and the hatching rate was measured for isolated groups of eggs of both species (F). For (A), n = 200 animals. For (B), n = 4325 animals. For (C), n = 200 animals. For (D), n = 679 animals. For (E), n = 10 animals. For (F), *D. laeve*, n = 438 clutches; *A. valentianus*, n = 84 clutches.

design allows easy navigation to the collections of both species. Upon entering each of them, controls at the top of the page allow navigation between species and to the different planes and sections (Fig 5C). On each of the numbered sections it is possible to turn on and off a marker

layer containing abbreviated and full names of the main structures and organs. On the bottom of the page a footer ribbon indicates the current section also allowing lateral navigation to other images and a slug ideogram at the bottom left corner indicates the approximate anatomical location of the section in view. Moreover, zooming control allows up to a ten-fold image magnification and sliding selection of the magnified region (Fig 5D). Image download options allow users to obtain full or cropped-up images. The phylogenetic analysis with Cox1 sequences of the species used to confirm the identity of both species in the atlas is also shown in the main page.

Manual tracing of the external contour of each section and segmentation of the major internal organs allowed the reconstruction of 3D models of both species (Figs 3D, 3D' and 4D, 4D'). These models can be explored on a 3D viewer accessed via the main SlugAtlas web page and allow the visualization of each of the organs independently of the rest or as part of the whole digital model (Fig 5E and S1 and S2 Videos). In the viewer, whole body 3D models are

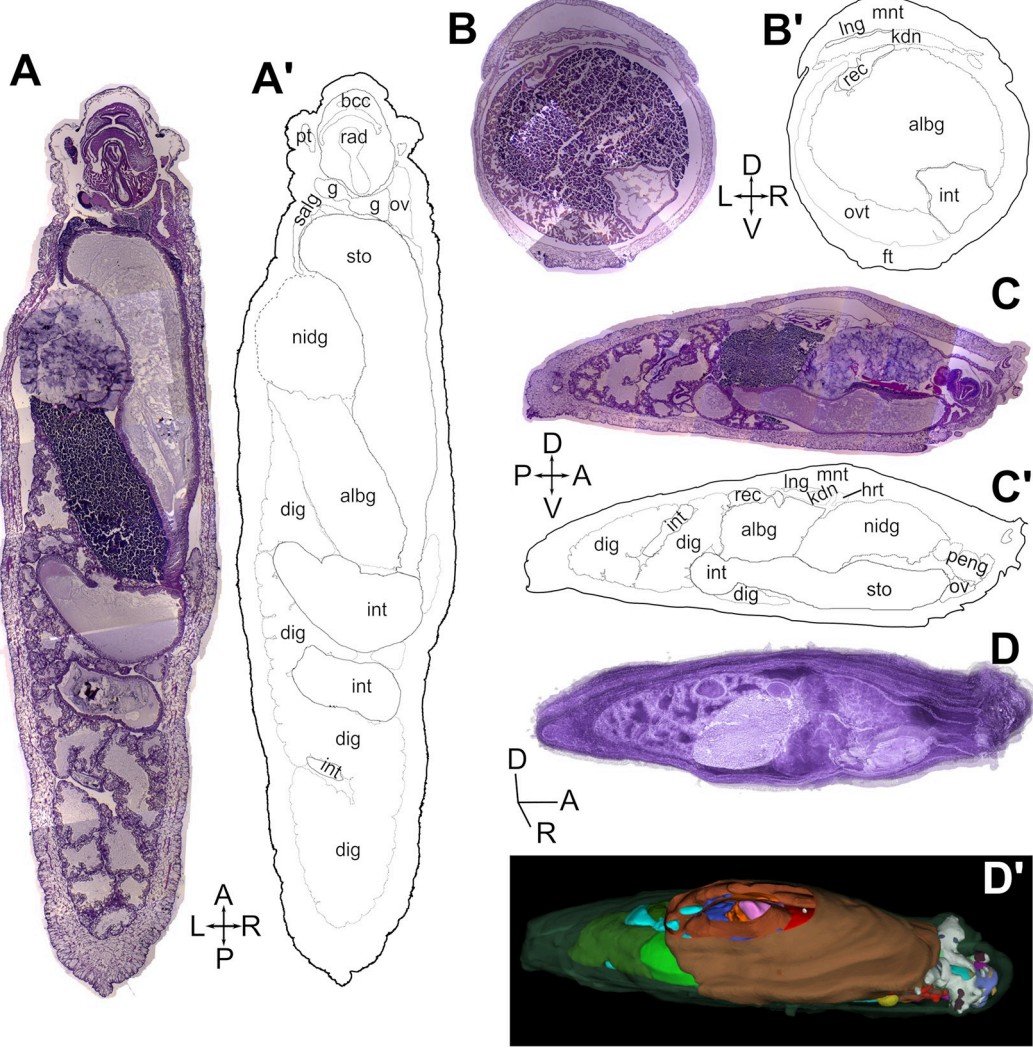

**Fig 3. Histology and anatomy of *D. laeve*.** Whole specimens were sectioned and stained with hematoxylin-eosin on the horizontal (A, A'), transversal (B, B'), or sagittal planes (C, C'). Each series was digitized, assembled in mosaic and layered into a digital stack (D). Manual tracing of the external contour of each section and segmentation of the major internal organs were used to build a 3D model (D, D'). Abbreviations in Table 1.

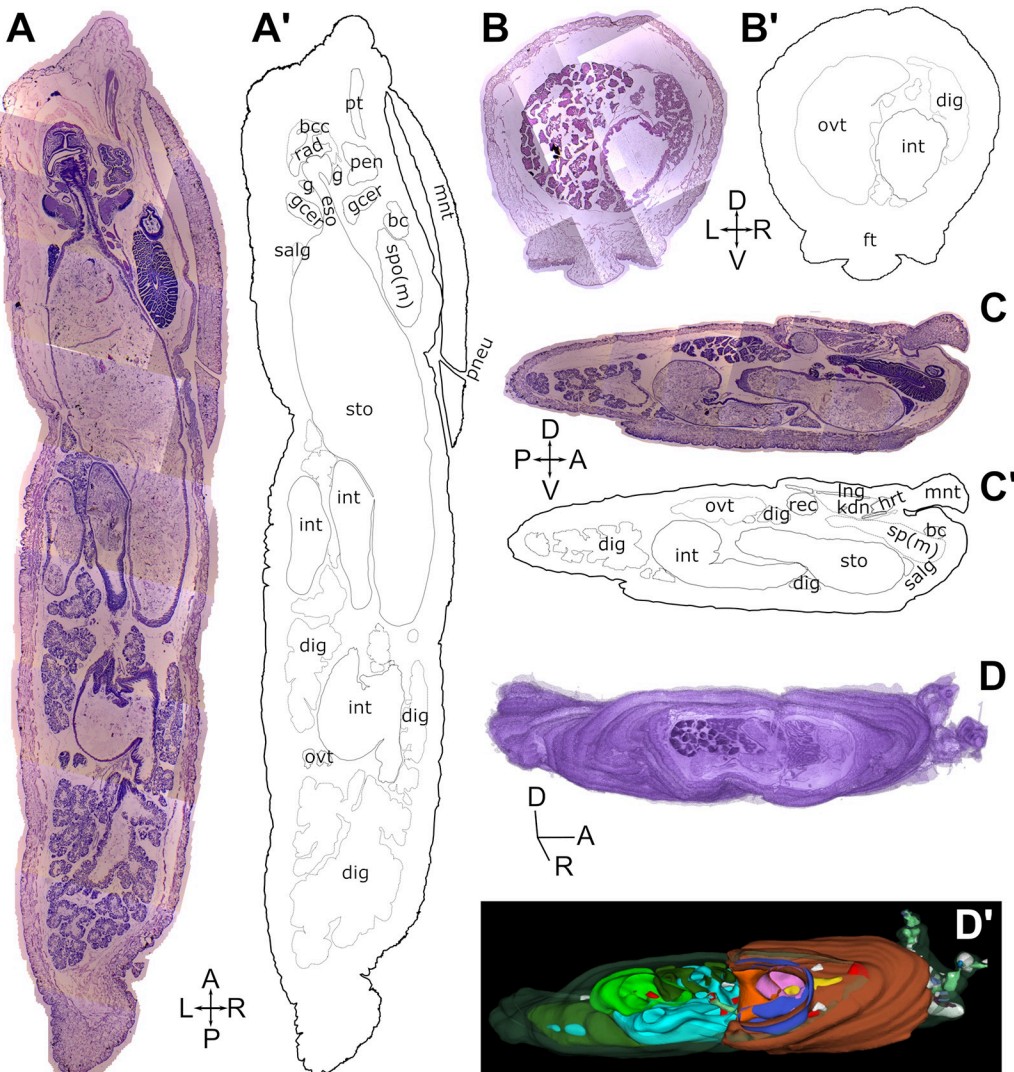

**Fig 4. Histology and anatomy of *A. valentianus*.** Whole specimens were sectioned and stained with hematoxylin-eosin on the horizontal (A, A'), transversal (B, B'), or sagittal planes (C, C'). Each series was digitized, assembled in mosaic and layered into a digital stack (D). Manual tracing of the external contour of each section and segmentation of the major internal organs were used to build a 3D model (D, D'). Abbreviations in Table 1.

available for both species and a separate one for the circumesophageal nerve ring of *D. laeve* which is an arrangement of nerve ganglia and nerve connectives around the esophagus (Fig 5E and 5F). Such 3D volume reconstructions allowed the visualization with high resolution of the relationships of organs not apparent from individual sections.

SlugAtlas also supports navigation on mobile devices with the main functionalities, albeit with a streamlined small-screen design. Furthermore, the atlas is designed to accommodate additional histological collections of different experimental conditions, and even of other species as they are generated in future work. Abbreviations for all figures are shown in Table 1.

## Histological comparison of *D. laeve* and *A. valentianus*

Comparison of the gross anatomy of these species revealed that they are similar with some clearly defined differences. Among them are differences in the lung-kidney-heart complex, the

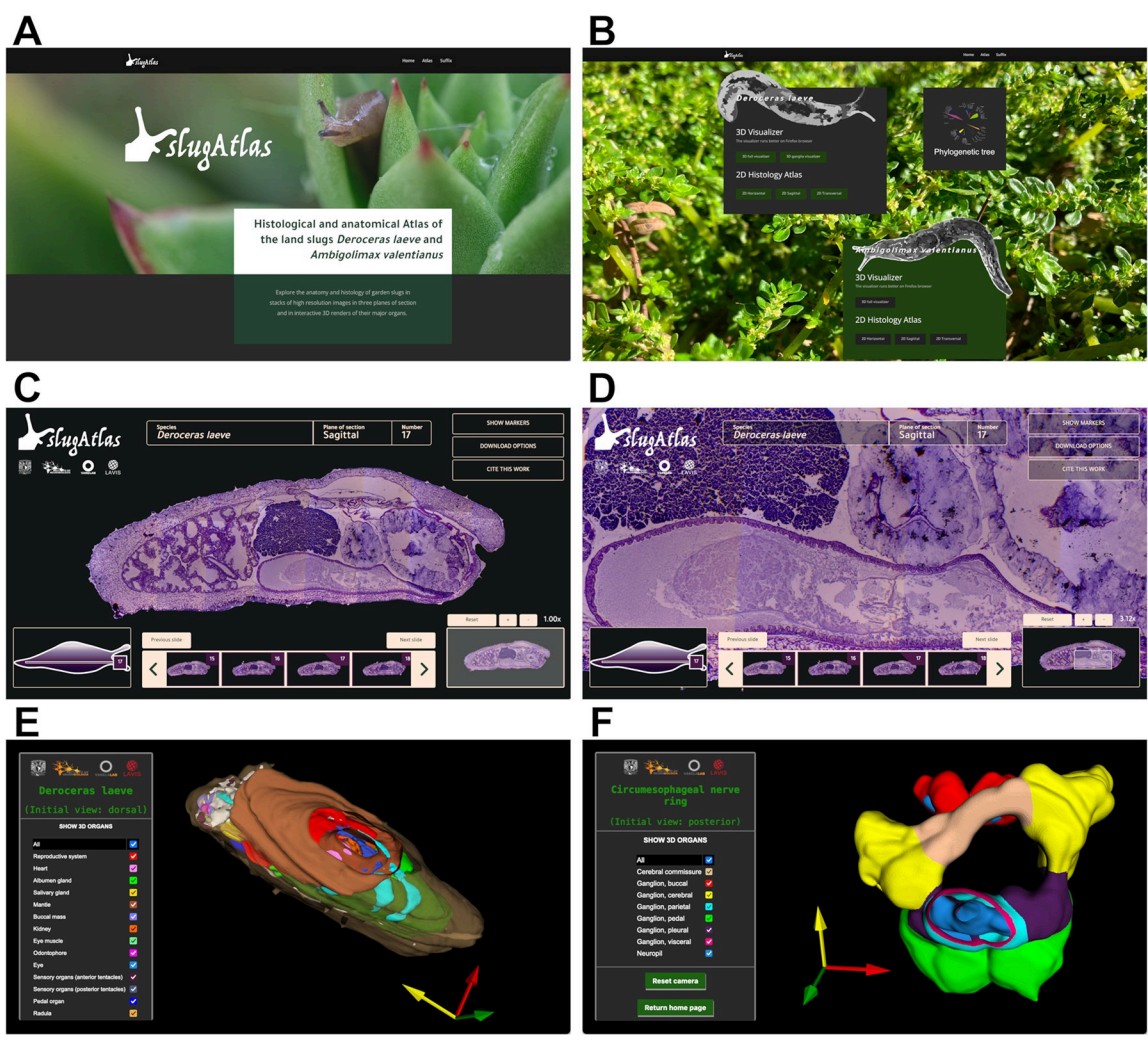

**Fig 5. Views of SlugAtlas.** Screenshots of the different sections of SlugAtlas (https://slugatlas.lavis.unam.mx) exemplify the various functionalities of the resource. (A) Landing page. (B) Second page with the buttons to access the 3D visualizer and the 2D histology collections. (C) Example of histological section of *D. laeve* and (D) a magnified view of a region of that section. (E) Example of 3D model view of the whole body and (F) of the circumesophageal nerve ring of *D. laeve*.

location and relative size of the ovotestis, and in *A. valentianus*, differences in the sexual organs between different stages of life. These differences are described in more detail in the following sections which address the histological features of the major organs and tissues of both species with emphasis in *D. laeve*. The images obtained from SlugAtlas are complemented in this work with images obtained of preparations made with other histological methods such as Kluver-Barrera or silver stain, or semithin sections stained with toludine blue.

## Body wall and mantle

The outer surface of the body wall or integument is covered by the epidermis which is a single layer of columnar epithelial cells interspersed with the outer ends of mucus secretory cells of various types and pigmented cells (Fig 6A–6F, 6I–6M). Underlying the epidermis is the sub-epidermal connective (sec) [29] which consists of a mesh of connective tissue and at its deep boundary contains bundles of muscular cells that in the dorsal region of the animal form a

**Table 1. Abbreviations used in this work.**

| | | | |
|---|---|---|---|
| albg | albumen gland | mndl | mandible |
| an | anus | mnt | mantle |
| aor | aorta | mntc | mantle cavity |
| apa | artery, pedal anterior | mt | myocyte trabeculae |
| app | artery, pedal paired | nidg | nidamental gland / oviducal gland |
| appo | artery, pedal posterior | npl | neuropil |
| ar | airway | nrv | nerve |
| at | anterior tentacle | odnt | odontophore |
| atdgo | anterior tentacle digitate organ | olfn | olfactory nerve |
| atn | anterior tentacle nerve | olforg | olfactory organ |
| atosh | anterior tentacle outer sheath | omsc | odontophore muscle |
| atr | heart atrium | opn | optic nerve |
| atrm | anterior tentacle retractor muscle | ov | oviduct / free oviduct |
| atse | anterior tentacle sensory epithelium | ovt | ovotestis |
| atsn | anterior tentacle sensory nerve | ovt(h) | ovotestis, hermaphrodite |
| bc | bursa copulatrix / spermatheca | ovt(m) | ovotestis, predominantly male |
| bcc | buccal cavity | pbc | peduncle of bursa copulatrix |
| bcm | buccal mass | pbg | pre-buccal groove |
| cc | collar cells | pcar | pericardium |
| ccom | cerebral commissure | pcl | pigmented cell layer |
| cepd | ciliated epidermis | pdcoc | peduncle, ocular |
| cpcon | cerebropedal connective | pedg | pedal gland |
| cplcon | cerebropleural connective | pen | penis |
| cutc | cuticle | penc | penial complex |
| cx | ganglionar cortex | peng | penial gland |
| dig | digestive gland / hepatopancreas | phar | pharynx |
| epd | epidermis | pneu | pneumostome |
| eso | esophagus | pr | prostate |
| eye | eye | pt | posterior tentacle |
| f | female duct of spermoviduct | ptdgo | posterior tentacle digitate organ |
| fb | fibroblastoid cells | ptish | posterior tentacle inner sheath / retractor muscle |
| fmsc | free muscle | ptosh | posterior tentacle outer sheath |
| ft | foot | ptse | posterior tentacle sensory epithelium |
| g | ganglion | rad | radula |
| gat | ganglion, anterior tentacle | radsc | radular sac |
| gatr | genital atrium | rec | rectum |
| gbuc | ganglion, buccal | ret | retina |
| gc | glandular cell | rmsc | radular muscle |
| gcer | ganglion, cerebral | salg | salivary gland |
| gnd | gonad | sec | subepidermal connective |
| gnp | gonopore / porus genitalis | sg | secretory gland |
| gpar | ganglion, parietal | sh | shell |
| gped | ganglion, pedal | shg | shell gland |
| gple | ganglion, pleural | So | Semper´s organ |
| gpt | ganglion, posterior tentacle | So(g) | Semper´s organ (glandular component) |
| gSo | ganglion, Semper´s organ | So(s) | Semper´s organ (sensory glomeruli) |
| gvis | ganglion, visceral | So(se) | Semper´s organ (sensory epithelium) |
| h | hemocyte | sol | sole |

(*Continued*)

**Table 1.** (Continued)

| | | | |
|---|---|---|---|
| hd | hermaphroditic duct | spd | spermatic duct, spermiduct |
| hrt | heart | sper | spermatophore |
| hs | hemolymphatic sinus | spo | spermoviduct |
| hv | hemolymphatic vessel | spo(f) | spermoviduct, female |
| int | intestine | spo(h) | spermoviduct, hermaphrodite |
| jaw | jaw | spo(m) | spermoviduct, male / prostate duct |
| kdn | kidney | stat | statocyst |
| Lc | Leydig cell | sto | stomach |
| lng | lung | t | tentacle |
| lngc | lung cavity | ur | ureter |
| lns | lens | vd | vas deferens |
| m | male duct of spermoviduct | vsem | vesicula seminalis |
| mcb | muscle cell bundle | vt | heart ventricle |
| mcl | muscle cell layer | | |

Abbreviations in this table also correspond to those of the SlugAtlas web page.

layer of a complex mesh (Fig 6A, 6I, 6G, 6H, 6N and 6O). The deeper ends of the secretory mucus cells of the epidermis extend into the external aspect of the subepidermal region. Deeper, beyond the muscular mesh is a loose tissue that coalesces with the rest of the body wall with no clear morphological distinctive features. This region is directly irrigated by the

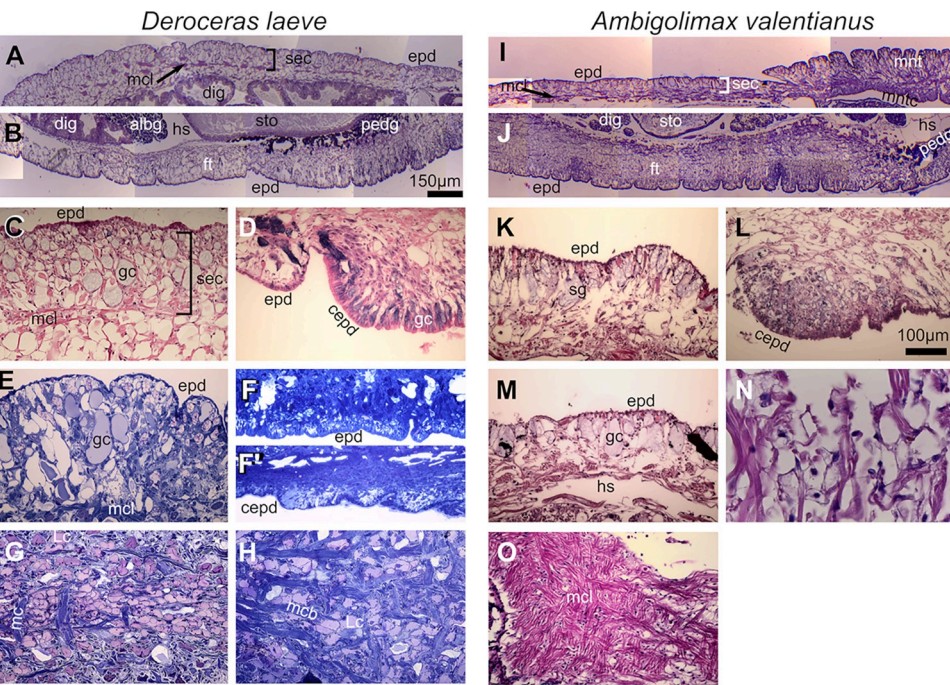

**Fig 6. Body wall of *D. laeve* and *A. valentianus*.** (A, C, I, and K) Dorsal or (B, J) ventral body wall in longitudinal sections. (D and L) Lateral fold of the foot in transverse section. (E and M) Epidermis of the mantle in transverse section (E is a semithin section). (F and F') Semithin longitudinal sections of the foot. (G-H) Semithin sections of the sec in horizontal plane. (N) Muscle bundles of the subepidermal connective (sec) in horizontal plane. (O) Muscular mesh in the subepidermal connective of the mantle in the horizontal plane. Scale bar in B applies to A, B, F, F', I, J, and O; bar in L applies to C-E, G, H, and K-N. Abbreviations in Table 1.

hemolymph contained in sinuses which expand and contract as the animal moves. Numerous cells similar to the mucus-producing Leydig cells observed in prosobranchs [30] are scattered throughout the subepidermal connective as can be best observed in semithin sections (Fig 6G and 6H). The integument varies in different regions of the animal such as the dorsal, lateral and pedal regions. The dorsal epidermis contains a larger proportion of secretory cells and microvilli (Fig 6A, 6C, 6E, 6I, 6K and 6M) and specific regions of the epidermis of the foot contain ciliated cells [31] (Fig 6D, 6F' and 6L).

The mantle, the structure that covers the visceral mass, is located over the mantle cavity that contains the shell and under it, the lung-kidney-heart complex. On its anterior region, the mantle is detached from the body forming the free mantle. The epidermis on its dorsal side is similar to the dorsal covering of the body and its ventral side is largely devoid of glands as is the dorsal body wall located under the free mantle (not shown). The pneumostome, or respiratory opening, is located on the right side of the mantle (Fig 4A and 4A') and is covered by ciliated epithelium. Just anterior to the pneumostome, the excretory slit is located.

No major differences were observed in the epidermis and subepidermal connective of both species except for the stronger pigmentation of epidermal cells in *A. valentianus*.

Concretions, which are particulate material found within tissues of gastropods, were also found within the subepidermal connective in both species identified by a refringent appearance with DIC optics (S2 Fig). In *D. laeve*, rosette shaped and bilobed crystals were observed with an approximate size range from 6 to 20 μm (S2A, S2A', S2B and S2B' Fig). In *A. valentianus*, smaller concretions were observed forming amorphous aggregates (S2H and S2H' Fig). Isolation of these concretions from *D. laeve* revealed that they are heat resistant, confirmed their bilobed and round shape and transmission electron microscopy revealed in cross-section an inner structure of concentric rings suggesting their growth by accretion (S2C and S2D Fig). Energy dispersed X-ray spectroscopy analysis (S3A Fig and S2 Table) revealed that these concretions contain predominantly oxygen (44.47%), carbon (12.17%), phosphate (1.76%), potassium (0.64%) and calcium (1.96%). An additional striking finding, however, was their high content of manganese (39%).

## Foot and suprapedal gland

The foot is divided in three main longitudinal sections separated by superficial grooves more readily observable in *D. laeve* (Fig 7B, 7C, 7I and 7J). Moreover, the epidermis of the foot is rich in glandular cells [31] which appear to be larger and with a lower density in *D. laeve* than in *A. valentianus* and longitudinal ciliated bands are present in the outer edges of the lateral sections of the foot in both species (Fig 6D, 6F' and 6L). Deeper, muscle cell bundles are surrounded by connective tissue and dotted with diverse types of hemocytes (Fig 6G and 6H). F-actin staining with fluorescent phalloidin reveals the mesh of muscle bundles located deep in the foot that are involved in body contraction (Fig 8).

The foot also contains the suprapedal gland which secretes mucus involved in locomotion [32] and extends from the ventral anterior region of the animal to approximately the caudal end of the stomach (Fig 7). This gland consists of a central duct that runs longitudinally (asterisks) and an associated network of secretory cells with densely basophilic cytoplasm forming what appear to be reticular transport streams (white arrows in Fig 7D–7F and 7K–7M). The suprapedal gland duct drains on its anterior end and into the prebuccal groove (Fig 7G and 7N) [32]. The roof of the duct is formed by a single layer of squamous or cuboidal cells and the ventral epithelium is composed of a single layer of cylindrical ciliated cells flanking a ventral groove to which the network of secretory streams appear to be oriented and where the basophilic mucus secretion can be observed. The anterior end of the duct and the prebuccal groove region is lined by ciliated epithelium.

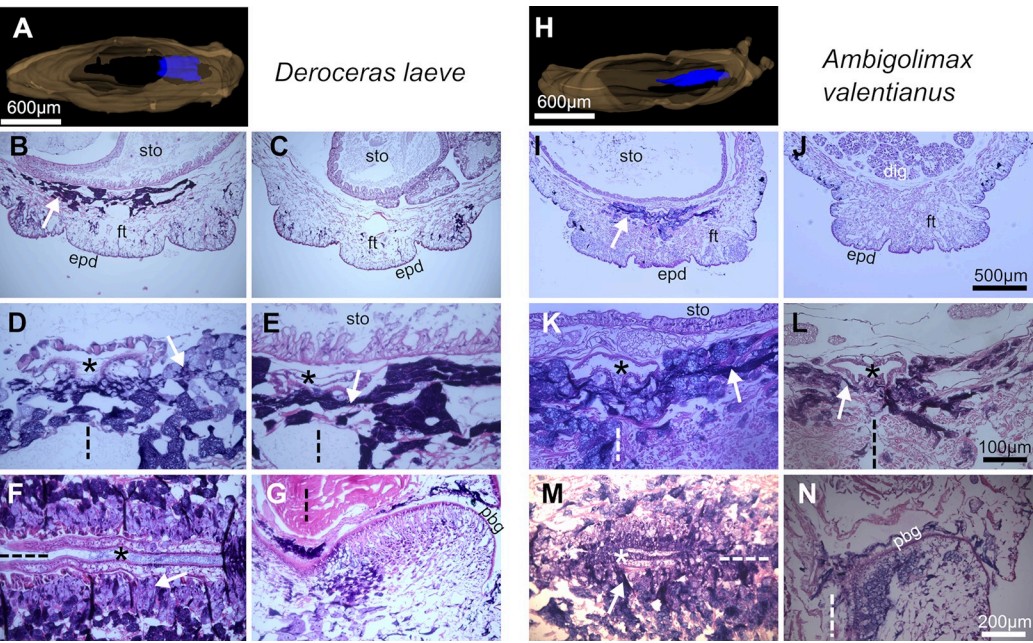

**Fig 7. Foot and suprapedal gland of *D. laeve* and *A. valentianus*.** (A and H) 3D models of the whole body (brown) and the suprapedal gland (blue). (B and I) Transverse sections of the foot in anterior and (C and J) in posterior regions (arrows indicate suprapedal gland). (D and K) Transverse sections of the suprapedal gland in anterior and (E and L) in posterior regions. (F and M) Horizontal sections on the posterior and (G and N) anterior regions of the suprapedal gland. Asterisks indicate the lumen of the suprapedal gland, dotted lines indicate the anatomical midline, and arrows in D-M indicate secretory streaks of the suprapedal gland. Scale bar in J applies to B, C, and I. Bar in L applies to D, E, and K. Bar in N applies to F, G, and M. Abbreviations in Table 1.

## Tentacles and Semper's organ

The anterior tentacle of *D. laeve* and *A. valentianus* is also called ommatophore as it bears the photosensory organ. In its everted configuration, this tentacle consists of an outer dermo-muscular sheath which is continuous with the cephalic integument and an inner muscular sheath, the tubular retractor muscle that joins the outer tube at the distal tip of the tentacle where the sensory structures are located (Fig 9A) [8, 33]. This inner sheath is covered by pigmented cells on the inside and on the outside. The outer sheath contains a chemosensory ciliated plate on its ventral tip which is covered by a dense mucus (not shown). In the inside distal region of the tentacle, the eye cup is located adjacent to the digitate organ which contains olfactory glomeruli and is continuous with the tentacular ganglion. From the eye cup and the ganglion, the optic nerve and the olfactory nerve emerge, respectively, running through the lumen of the inner sheath towards the cerebral ganglia of the circumesophageal nerve ring. At the distal region of the inner sheath lumen, clusters of round large cells with granular cytoplasm, the collar cells, are loosely attached to the muscle fibers via thin cell projections. Upon retraction of the tentacle to the head of the animal by effect of the retractor muscle, the outer sheath turns outside in and the sensory epithelium is now located inside the inverted sheath which is continuous with the exterior (Fig 9B and 9C). This is the form we studied and describe herein as it is the configuration that results upon anesthesia and tissue fixation for histological processing.

An asymmetry was observed in the retracted posterior tentacles when viewed from the top as the right retractor muscle is mostly straight and the left is curved medially on its caudal end,

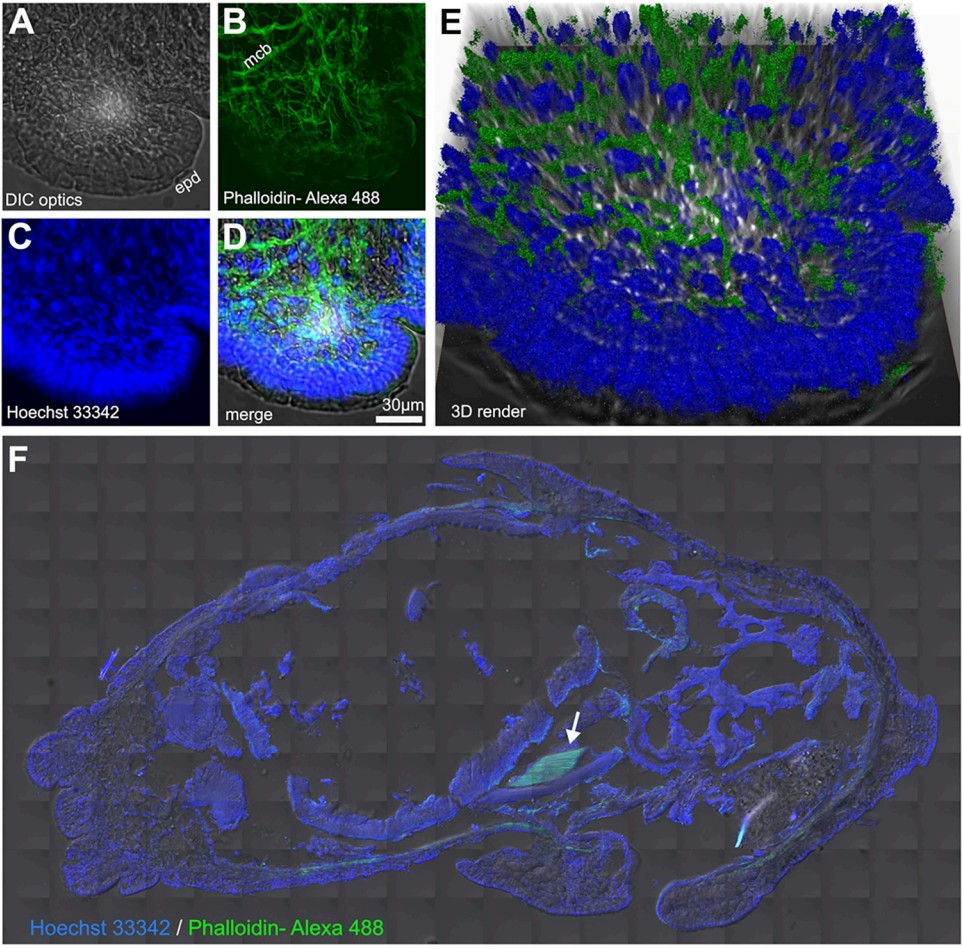

**Fig 8. Muscle bundles of *D. laeve* visualized with phalloidin staining.** (A-E) Lateral pedal fold in a transverse section showing (A) tissue texture obtained by DIC optics, (B) phalloidin staining, (C) Hoechst 33342 (nuclei), (D) 3-channel projection, and (E) 3D rendering. (F) Horizontal section showing the free muscle stained with Phalloidin (arrow) with Hoechst 33342 counterstain. Abbreviations in Table 1.

both merging into the free muscle that runs along the reproductive organs to the dorsal aspect of the animal (Fig 8F). In both tentacles, however, the eye cup is laterad, close to the outer body wall and the digitate organ and ganglion are located on their medial side. The eye cup contains a cornea, a lens, a retina and an outer pigmented epithelium. No major differences were observed between *D. laeve* and *A. valentianus* except in the larger volume occupied by the different components of the tentacle in the latter (Fig 9C, 9E, 9F and 9I).

The anterior tentacle, the rhinophore, is shorter and also contains an outer dermo-muscular sheath which is continuous with the integument, but the retractor muscle does not form a tube. It is instead a band parallel to the tentacular nerve which on its anterior end splits into smaller bands that anchor to the inner side of the outer sheath (Fig 9G, 9H and 9J, 9K) and caudally joins the posterior retractor muscles forming the free muscle. Retraction of this muscle pulls and inverts the outer sheath to the head region (diagram in Fig 9D). This tentacle contains a digitate organ and its attached ganglion which is flanked by cells similar to the collar cells of the posterior tentacles (Fig 9G, 9H, and 9J, 9K). The ganglion projects a sensory nerve that runs adjacent to the retractor muscle, merges with the

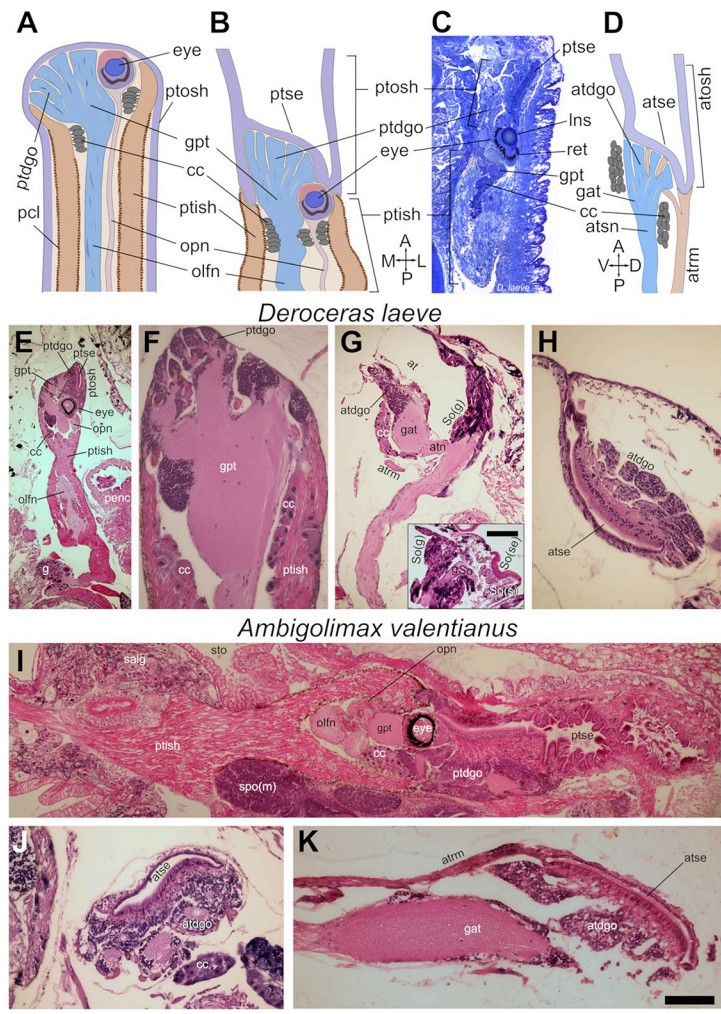

**Fig 9. Tentacles and Semper's organ of *D. laeve* and *A. valentianus*.** (A) Diagram of posterior tentacle in everted and (B) in retracted configuration. (C) Horizontal semithin section of the right posterior tentacle of *D. laeve*. (D) Diagram of anterior tentacle in retracted configuration. (E-F) Posterior and (G-H) anterior tentacle and Semper's organ (So) of *D. laeve*. Inset in (G), Semper's organ in a more lateral position. (I) Posterior and (J-K) anterior tentacle of *A. valentianus*. Values of scale bar: E (400 µm); F-H and J (100 µm); C and E (300 µm); inset in G (300 µm); and K (80 µm). Diagram in (A), modified from [33]. Image in (I) was assembled in mosaic manually from several micrographs of the same section in Gimp. Abbreviations in Table 1.

nerve of Semper's organ (see description below) (Fig 9G) [34], and projects centrally to the cerebral ganglion (not shown).

The Semper's organ is a dual sensory-glandular structure that contains a ganglion connected to a subepidermal sensory component with a glomerular organization akin to that of the tentacular digitiform ganglia (Fig 9G and inset) [34]. This subepidermal element is apposed to a ciliated epidermal plate facing the ventral side of the animal, just anterior to the prebuccal groove. The ganglion of the Semper's organ is surrounded by a cluster of cells with glandular appearance similar to those of the collar cells and with basophilic secretory streams resembling those of the suprapedal gland that terminate in the epidermis, anterior and posterior to the ciliated plate. This organ was also observed in *A. valentianus* and no major differences were detected in anterior tentacles between both species.

## Nervous system

The circumesophageal nerve ring is composed of cerebral, parietal, pleural, visceral, and pedal ganglia that project nerves into different body regions (Fig 10A–10C, and 3D printed model in Fig 10K). Moreover, buccal ganglia are located close to the nerve ring in the caudal aspect of the buccal complex (Fig 10G and 10J). Ring ganglia contain several cell types in their cortical regions surrounding a central neuropil traversed by axonal fibers for the most part devoid of cells (Fig 10D–10J). Cells of several sizes are located in the cortex of the ganglia and include mostly neurons as well as support or glial-type cells. Ganglia display cells ranging from small with scantly cytoplasm as in the cerebral and pedal ganglia to large with clearly defined nuclei and nucleoli as in the pleural and buccal ganglia. Many of the large cells reveal the proximal end of their axon projecting into the neuropil. Dorsal and ventral commissures and ipsilateral nerve connectives are mainly composed of axon fibers and contain scattered interfascicular nerve cells (Fig 10H). The outer surface of the nerve ring is covered by a thin perineural layer containing pigmented cells. No significant anatomical differences were detected between the nervous system of *D. laeve* and *A. valentianus*.

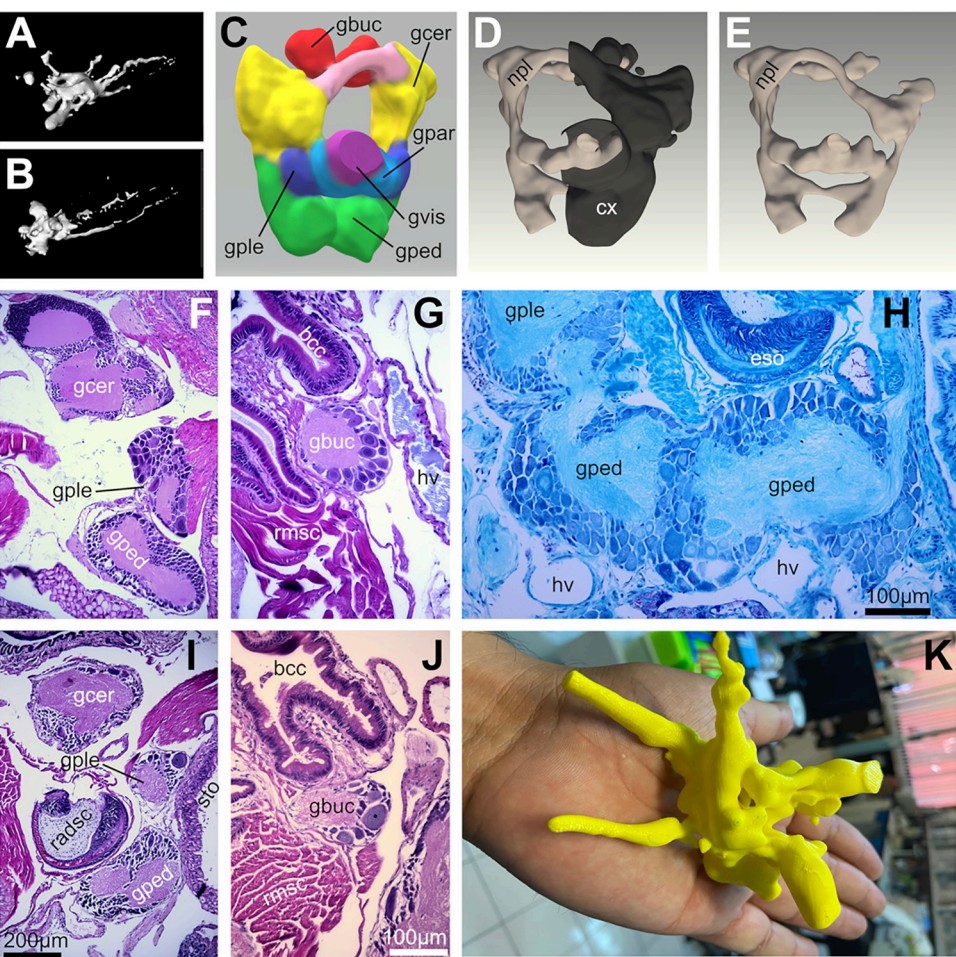

**Fig 10. Nervous system of *D. laeve* and *A. valentianus*.** (A) 3D render of *D. laeve* and (B) of *A. valentianus*. (C) 3D model of the circumesophageal nerve ring of *D. laeve* showing the ganglia in different colors, and in D and E, the cellular cortical regions in dark color and the neuropil in light color. (F-G) Sagittal sections of *D. laeve* and (H) transverse section with Kluver-Barrera stain. (I-J) Sagittal sections of *A. valentianus*. (K) 3D-printed model of the nervous system of *D. laeve* for educational purposes. Scale bar in I applies to F. Bar in J applies to G. Abbreviations in Table 1.

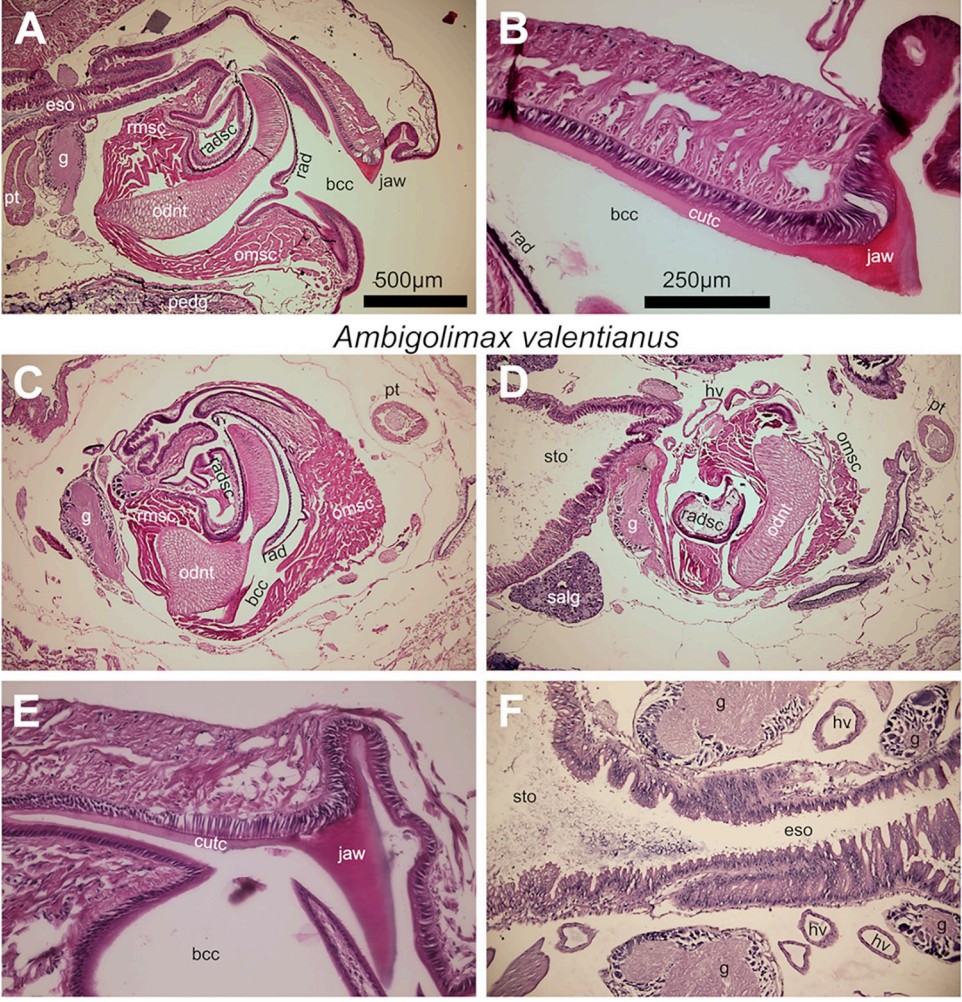

**Fig 11. Buccal organ of *D. laeve* and *A. valentianus*.** Sagittal sections are shown. (A) Medial region of buccal organ. (B) Dorsal lip with jaw and cuticle. (C) Medial region of buccal organ. (D) More lateral region than in C. (E) Dorsal lip with jaw and cuticle. (F) Esophagus. Scale bar in A applies to C and D. Bar in B applies to E and F. Abbreviations in Table 1.

## Digestive system

**Buccal complex.** The anterior-most structure in the buccal region is the jaw located on the dorsal lip of the mouth (Fig 11A, 11B and 11E). The jaw is a sharp chitinous structure used for cutting foodstuff which is secreted by an epithelium formed by a single layer of columnar cells with parabasal nuclei and apical vesicles (Fig 11B and 11E). The jaw is continuous with the thin chitinous cuticle that lines the foregut [35] and connects to the also chitinous radula, the rasping, tongue-like structure characteristic of several mollusk Classes including gastropods. The complex formed by the jaw, cuticle, and radula is a single unit resistant to SDS and proteinase K treatment (not shown). Just over the columnar epithelium of the jaw and cuticle, a heavily muscular structure gives motility to the jaw and upper lip during feeding. The radula extends from a ventral fold just inside the mouth and surrounds the cartilaginous odontophore extending into the odontogenic organ, the radular sac [36] (Fig 11A, 11C and 11D). The radula has its denticles exposed to the oral cavity (Fig 11A and 11C) and it lies over the subradular

epithelium, which is non-stratified and formed by short cylindrical cells with basal nuclei that thins out from the lip and into the oral cavity to squamous cells and thickens again and becomes stratified upon reaching the odontogenic region. This epithelium is located over the odontophore which is in turn formed by a pseudostratified layer of long columnar chondrocytes and gives support to the radula. In the radular sac, the radula curls around a core that is muscular at its medial aspect and filled with fibroblastoid cells at its lateral regions (Fig 11A, 11C and 11D). The ventral and anterior aspect of the odontophore is supported by the thick odontophore muscle and joining its caudal and dorsal end with the radular organ is the radular muscle. The buccal cavity continues caudally into the esophagus (Fig 11A and 11F) turning form a columnar non-stratified epithelium with parabasal nuclei into the pseudostratified columnar epithelium with villi, characteristic of the stomach. The main buccal difference between *D. laeve* and *A. valentianus* is that the jaw of the former is thicker and shorter, compared to a thinner and longer in the latter (Fig 11B and 11E, respectively).

**Stomach.** The stomach is a sac-like chamber located between the esophagus and the intestine characterized by a non-ciliated pseudostratified columnar epithelium with parabasal nuclei surrounded by a muscular layer (Fig 12A–12C and 12N–12P). In both species, columnar cells form villi that line the stomach and are short in the anterior part and longer in the caudal region near the intestine forming deeper crypts. Also in both species, extracellular vesicles are shed from the columnar cells into the stomach lumen which appear to be filled with smaller vesicular structures and range in size from 5 to 10 μm (Fig 12C, 12K and 12O). Although these species have been described to contain a crop and a stomach [12], no overt distinction of these regions can be made. A basal lamina covers the external aspect of the stomach and salivary glands are located on its anterior region (Fig 12H and 12U). The columnar cells and villi are shorter in *A. valentianus* than in *D. laeve*.

**Intestine.** At the transition between stomach and transverse intestine in which the digestive gland drains into the gut, the epithelium is also pseudostratified and columnar with cilia in their apical surface (Fig 12I, 12J and 12V). In the descending intestine the villi are not present and is formed by a ciliated columnar epithelium (Fig 12E, 12Q and 12R). In the ascending intestine the ciliated columnar cells are gradually shorter (Fig 12F and 12S) and at its terminal region near the anus the gut lining is made of cuboidal non-ciliated cells (Fig 12G and 12T). The intestine is for the most part surrounded by the digestive gland.

**Digestive gland.** The digestive gland is formed by secretory acini each surrounding a cavity with a proteinaceous appearance which coalesce forming lobules (Fig 12L, 12M and 12W). Acini are connected to a central lumen that drains to the gut at the transition region between stomach and intestine (Fig 12I, 12J and 12V). Acini are formed by holocrine-type cells of several sizes and connective tissue is found in the interstitial regions between acini and lobes, likely maintaining their structure. Digestive gland acini and lobules in *D. laeve* are smaller than in *A. valentianus*.

## Lung-kidney-heart complex

**Heart.** The heart of juvenile individuals of both species is located dorsally (Fig 13A and 13G) and in the anterior region of the mantle complex composed of an atrium and a ventricle with an atrio-ventricular wall surrounded by the pericardium which is a non-stratified squamous epithelium (Fig 13B and 13H). The myocardium is a contractile thin non-stratified cell layer (arrows in Fig 13D and 13J) and trabeculae of muscle cells traverse the periphery of the heart cavities and anchor to the myocardium, the trabeculae forming partial subdivisions of the atrium and being more abundant in the ventricle. The atrium is separated from the ventricle by muscular flaps that do not form a clearly discernible valve (Fig 13B, 13C, 13H and 13I).

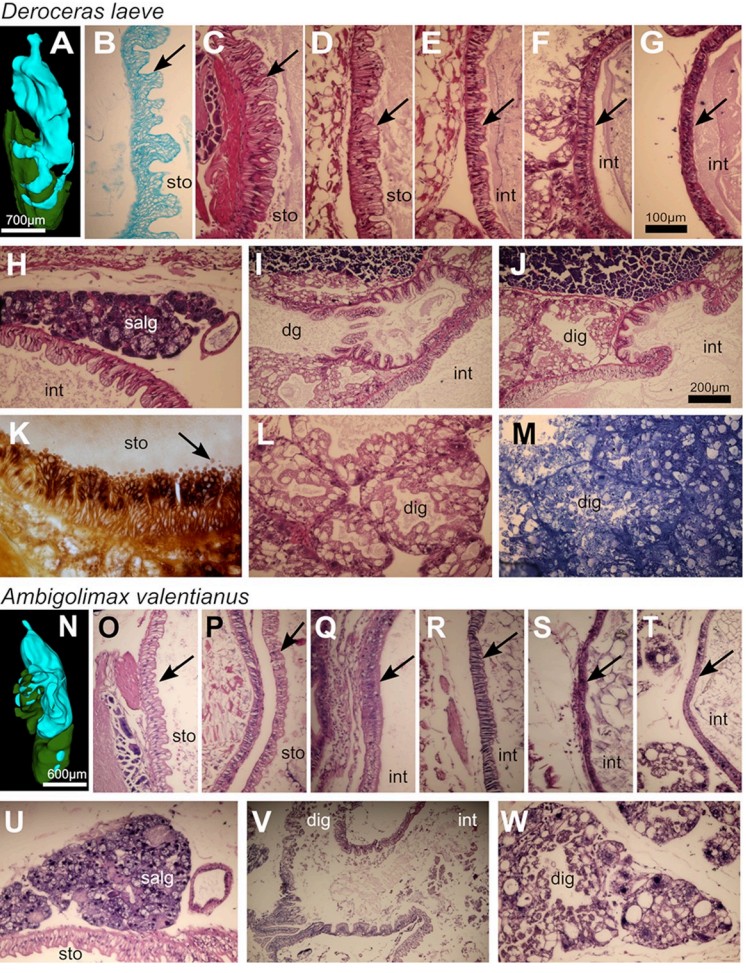

**Fig 12. Digestive system of *D. laeve* and *A. valentianus*.** (A-M) *D. laeve*. (A) 3D model of digestive system (digestive tube in cyan and digestive gland in green). (B, C, and K) Epithelium of the anterior region of the stomach (B is a Kluver-Barrera stain and K is silver stained). (D) Posterior region of the stomach. (E) Descending intestine. (F) Ascending intestine. (G) Ascending intestine near the anus. (H) Salivary gland. (I and J) Region where the digestive gland drains into the intestine. (L and M) Digestive gland (M is a semithin section). (N-W) *A. valentianus*. (N) 3D model of digestive system (digestive tube in cyan and digestive gland in green). (O) Epithelium of the anterior region of the stomach. (P) Posterior region of the stomach. (Q-R) Descending intestine. (S) Ascending intestine. (T) Ascending intestine near the anus. (U) Salivary gland. (V) Region where the digestive gland drains into the intestine. (W) Digestive gland. Arrows in B-G and O-T indicate the digestive tube epithelium. Scale bar in G applies to B-H, K-M, and O-U and W. Bar in J applies to I and V. Abbreviations in Table 1.

The heart occupies a larger relative volume in *A. valentianus* and the myocardium is not as thin as in *D. laeve*. Furthermore, in older individuals of both species the ventricle wall is thicker and more muscular and an atrio-ventricular semilunar valve is clearly present while the atrium remains thin-walled (Fig 13E, 13F and arrows in 13K, 13L).

**Lung.** In both species the lung forms a complex hollow structure with its largest curved area located dorsal to the heart and kidney complex and is distinctly separated from the mantle cavity by the mantle epithelium and a network of hemolymphatic sinuses (Fig 14A, 14B, 14D, 14E, 14G, 14K and 14L). The wall of the lung is a non-stratified cuboidal epithelium with apical nuclei and scattered mounds that protrude into the air cavity formed by thickening and infolding of the epithelium. Such mounds possess tufts of microvilli more clearly visible in semithin sections of *D. laeve* (Fig 14D, 14E and 14G). The dorsal side of the lung epithelium is

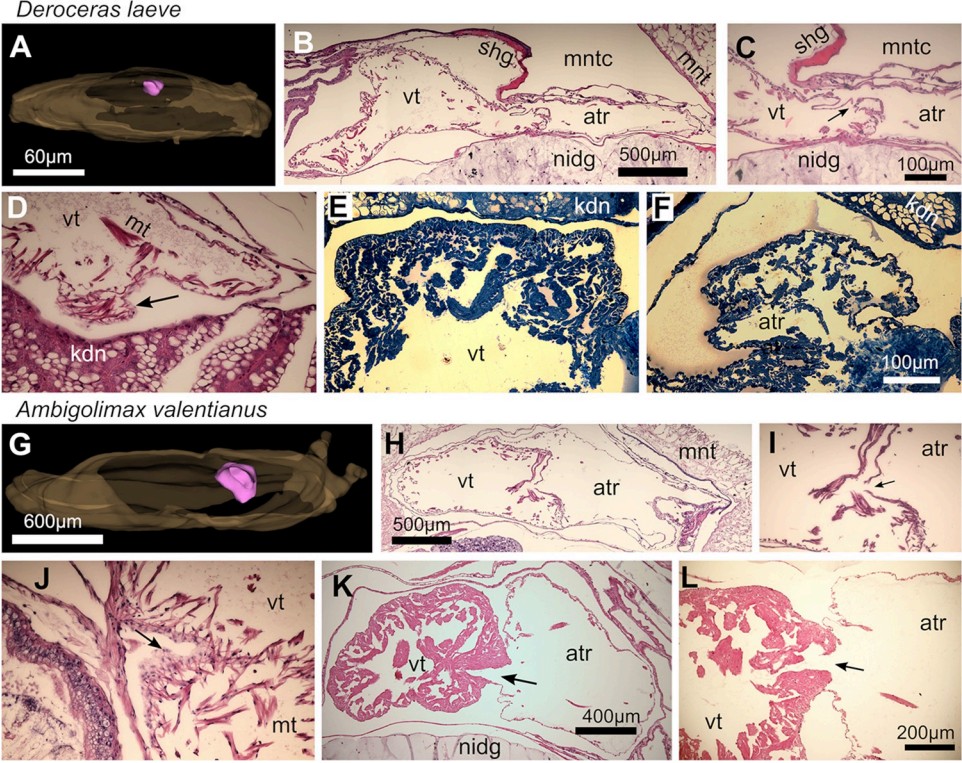

**Fig 13. Heart of *D. laeve* and *A. valentianus*.** (A-F) *D. laeve*. (A) 3D model of the whole body (brown) and the heart (pink). (B-D) Sections of the heart of a juvenile *D. laeve* and (E and F) of an older individual (semithin sections stained with toluidine blue). (G-L) *A. valentianus*. (G) 3D model of the whole body (brown) and the heart (pink). (H-J) Sections of the heart of a juvenile and (K-L) an older individual. Arrow in C and K-L indicates the atrioventricular valve, in D and J it indicates squamous cells of the myocardium. Scale bar in C applies to D-E. Bar in L applies to I-J. Abbreviations in Table 1.

lined by a mesh of muscle cells and further out, a layer of hemolymphatic vessels. From the caudal region of the gas exchange region of the lung, it continues to form an airway that travels anteriorly on the right side of the animal (not shown). The last region of this duct is lined by a non stratified squamous epithelium with ciliated cells that reaches up to the pneumostome. The lung and its airways are intimately associated with the kidney and ureter only separated by hemolymphatic sinuses located between them (Fig 14H and 14M).

 **Kidney.** The kidney in both species is a shell-shaped structure that encases or abuts the heart from its caudal side (Fig 14A and 14K). In *D. laeve*, the dorsal side of the kidney has rounded folds that are attached to the lung and have characteristic large hemolymphatic sinuses (Fig 14B). On its ventral side, it has lamellar folds projecting into the renal sac with thin hemolymphatic sinuses that appear to contain hemocytes (Fig 14C). Both types of folds are formed by a columnar non-stratified epithelium with large apical vacuoles containing concretions (see below) and basal nuclei interspersed with glandular cells. The hemolymphatic sinuses in the rounded folds typically contain proteinaceous and particulate material, while the basal sinuses of lamellar folds and the renal lumen, the renal sac, are mostly clear. The sinuses formed by the rounded folds and the kidney itself remain attached to the lung on its trajectory along the right side of the animal on its anterior course. In this region, the kidney turns into a tubular ureter that merges with the airway at its transition from a respiratory epithelium containing the previously described goblet cells [37] to the smoother airway lining (Fig 14H–14J). Renal lamellae in *A. valentianus* are longer, more branched, occupy a relatively larger cross

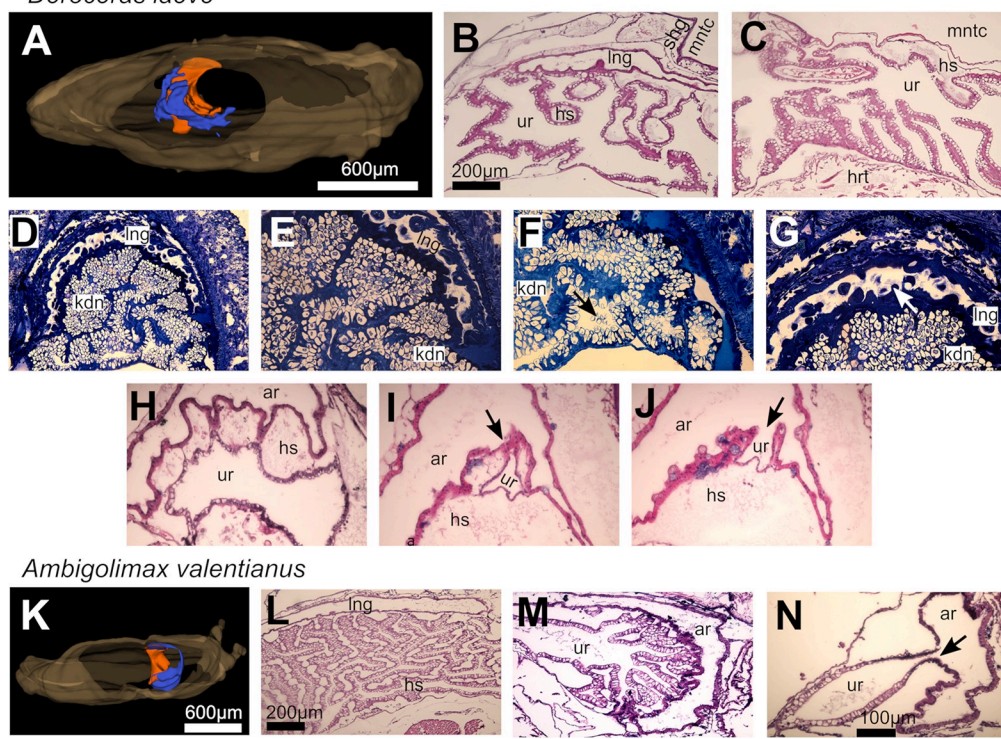

**Fig 14. Kidney and lung of *D. laeve* and *A. valentianus*.** (A-J) *D. laeve*. (A) 3D model of the whole body (brown), lung (blue), and kidney (orange). (B-C and H-J) Transverse sections of the lung and kidney. (D-G) Horizontal semithin sections. (H-J) Close interaction of the ureter with the airway in D. laeve. Arrow in F indicates renocytes bursting into the renal lumen releasing their concretions, in G it indicates the villi in the lung epithelium, and in I-J the merging of the ureter with the airway. (K-N) *A. valentianus*. (K) 3D model of the whole body (brown), lung (blue), and kidney (orange). (L-N) Transverse sections of the lung and kidney. (C and E) Close interaction of ureter with airway and their merging (arrow in N). Scale bar in B applies to C and D. Bar in N applies to E-J and M. Abbreviations in Table 1.

section, and have relatively larger hemolymphatic sinuses than in *D. laeve* (Fig 14L). Moreover, in contrast to *D. laeve*, the dorsal and ventral lamellae in *A. valentianus* are thin and no distinction can be made on rounded and lamellar folds. The interaction of the kidney with the lung is also present in *A. valentianus* but instead of ureter and airway running together, there is a branched region of the renal lamellae that is surrounded by the airway on the right region of the mantle complex (Fig 14M). The secondary ureter in *A. valentianus* also merges into the airway as in *D. laeve* (Fig 14N).

Macroscopic examination of the kidney reveals a white appearance which is due to concretions that are readily released from the tissue upon mechanical manipulation. Semithin sections of *D. laeve* allowed the best visualization of the concretions within the renocytes (Fig 14D–14G). These concretions are hollow and pleomorphic with rounded external surfaces and appear to reach the renal lumen upon bursting of the renocytes (Fig 14F and S1E and S1E' Fig). The concretions, which are heat labile, are composed mostly of nitrogen (37.72%), carbon (36.09%), and oxygen (25.07%), which is close to the expected stoichiometry of uric acid (34.2%, 36.7%, and 29.1% of the same elements, respectively)(S1G and S2B Figs and S2 Table). The renal concretions were also observed in the airway (not shown) and emanating from the excretory slit adjacent to the pneumostome as mucous masses of white appearance which suggests their excretory route.

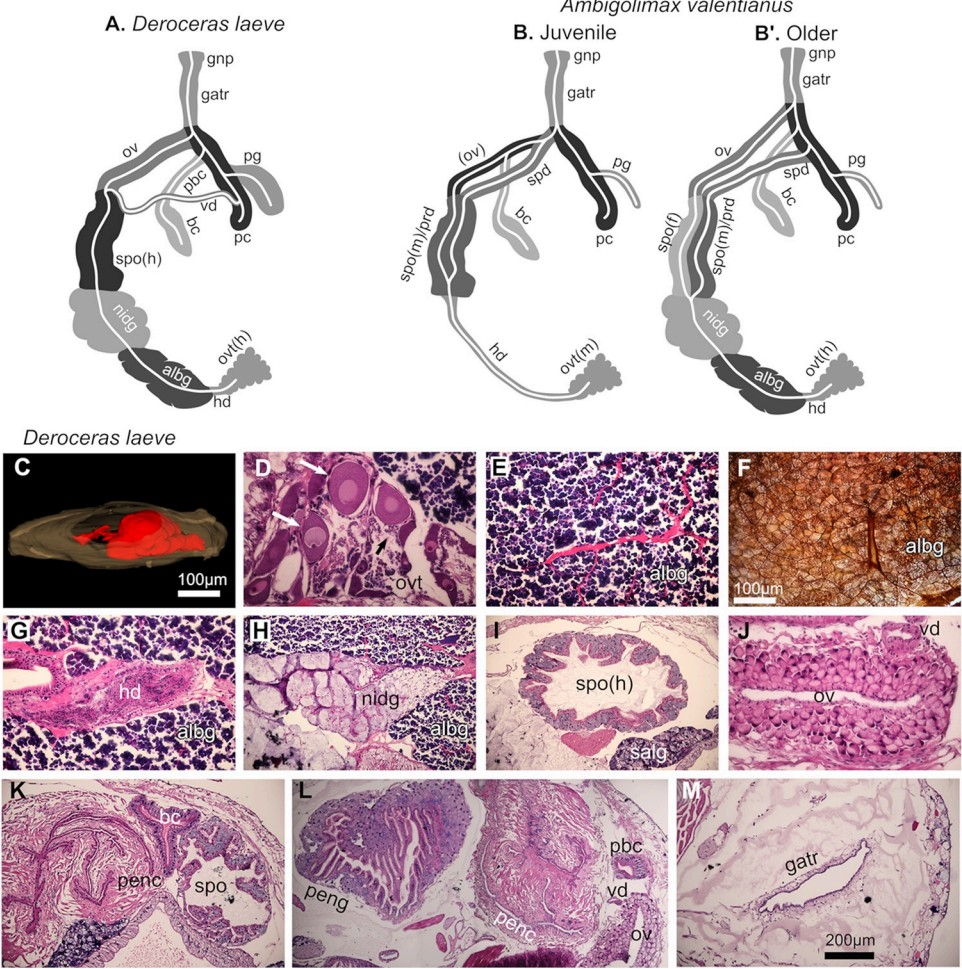

**Fig 15. Reproductive system of *D. laeve* and *A. valentianus*.** (A-B') Diagrams of both species show the topology of the different components but not their proportions. The topology differs between the two species and between the two ages analyzed of *A. valentianus*. White curved lines indicate the lumen of the different ducts that constitute the system and show their connection patterns. Modified from [38]. (C-M) Reproductive system of *D. laeve*. (C) 3D model showing the reproductive system in red. (D) ovotestis. (E) Albumen gland. (F) Albumen gland with silver stain. (G) Hermaphroditic duct and albumen gland. (H) Nidamental gland. (I) Spermoviduct. (J) Oviduct and vas deferens. (K) Penial complex, bursa copulatrix/spermatheca, and spermoviduct. (L) Penial gland, penial complex, peduncle of bursa copulatrix, vas deferens, and oviduct. (M) genital atrium. Scale bar in F applies to D, E, G, and J. Bar in M applies to H, I, K, and L. Abbreviations in Table 1.

## Reproductive system

Analysis of series of histological preparations of whole animals in the diverse planes of section allowed the tracing of all components and ducts of the reproductive system which is located dorsally and occupies a large proportion of the anterior region in both species (Figs 15C and 16A). This revealed topological differences between the two species and in *A. valentianus* between two ages analyzed (diagrams in Fig 15A, 15B and 15B'). The reproductive system of *D. laeve* revealed components consistent with hermaphroditic reproduction (Fig 15A) with the previously reported topology of phallic individuals [38]. In contrast, the juvenile *A. valentianus* had predominantly male features and the older individuals revealed hermaphroditic components (Fig 15B and 15B').

In *D. laeve*, the ovotestis or hermaphroditic gonad, is the caudal-most part of the reproductive system located deep in the digestive gland and is formed by two types of acini that produce either eggs or sperm cells (Fig 15A, 15C and 15D). The outer surface of the ovotestis has an appearance of bunch of grapes covered by a thin pigmented cell layer. Connected to the ovotestis, the hermaphrodite duct appears filled with sperm cells, it is lined by a single layer squamous epithelium, and on its anterior course runs through the albumen gland (Fig 15G). This gland is formed by dense lobes of cells with basophilic nuclei and unstained cytoplasm divided into lobules and traversed by collector ducts that coalesce into a major duct that drains into the nidamental gland (Fig 15E). Reduced silver staining revealed the honeycomb-like structure of the albumen gland allowing visualization of cell membranes, lobule divisions, and collector ducts (Fig 15F). The nidamental gland, also called oviducal gland, is composed of large lobes with diffuse and light basophilic staining and dotted by scattered cell nuclei and a network of fibroblastoid cells with small nuclei and eosinophilic cytoplasm (Fig 15H). The duct continues into the spermoviduct with folds that project into its lumen formed by large ovoid cells with clearly defined nuclei and basophilc cytoplasm and overlaid with a lining of ciliated cuboidal non-stratified epithelium (Fig 15I). One of the folds in the spermoviduct may act as sperm groove but it could not be distinguished based solely on morphology. A single squamous cell layer covers the outside of the spermoviduct. This transitions into the oviduct which has a lining of a cuboidal single cell layer surrounded by larger round cells with small nuclei located away from the lumen (Fig 15J). In transverse sections, the oviduct has a rosetta arrangement covered on its outer side by a loose muscular mesh. From the spermoviduct-oviduct transition region, a thin duct, the vas deferens, emerges and connects to a distal region of the penial complex (Fig 15A, 15J and 15L). The latter is a heavily muscular structure with a squamous non-stratified lining and a collapsed lumen forming complex folds with sharp angles that are stretched upon eversion of the penis (Fig 15K and 15L). The penial complex and oviduct merge and lead into the genital atrium or vestibule, whose lumen is lined by a non-stratified squamous epithelium, it is also surrounded by a loose mesh of muscle cells (Fig 15M), and leads to the genital pore. Between the connection of the oviduct with the penial complex and proximal to the connection with the vas deferens, two tubular structures emerge: the penial gland and, more proximal, the bursa copulatrix (Fig 15A and 15K, 15L). The penial gland is a robust structure with very complex luminal folds lined by a ciliated short columnar epithelium and surrounded by large round cells with basophilic cytoplasm (Fig 15L). The bursa copulatrix extends parallel to the penial complex reaching its caudal end and its lumen is lined by a ciliated short columnar non-stratified epithelium and covered by a single cell layer of squamous cells (Fig 15K).

In the juvenile *A. valentianus*, the ovotestis appears to be predominantly spermatogenic with only a few scattered ovogenic cells with basophilic cytoplasm, it occupies a larger volume that in *D. laeve*, it is more pigmented, and is located more dorsal (Figs 15B and 16B). The albumen and nidamental glands are not present, the hermaphroditic duct is filled with sperm cells (Fig 16C), and the spermoviduct at this age also constitutes the prostatic gland and contains characteristic secretory follicles (Fig 16E and 16F). The lumen of the spermoviduct splits into two ducts, hence becoming diaulic (Fig 16F). The male duct is lined by ciliated cuboidal cells, surrounded by a thick layer of muscle cells and receives the secretory contents of the prostatic follicles, while the female duct is not ciliated. The male duct leads to the sperm duct, and the female duct to the oviduct (Fig 15B) which does not have the rosetta appearance of the hermaphroditic oviduct of *D. laeve* (Fig 16G). Sperm duct and oviduct merge and then coalesce with the penial complex (Fig 16G and 16H), before leading to the genital atrium and gonopore (not shown). In further contrast with *D. laeve*, the juvenile *A. valentianus* does not have an

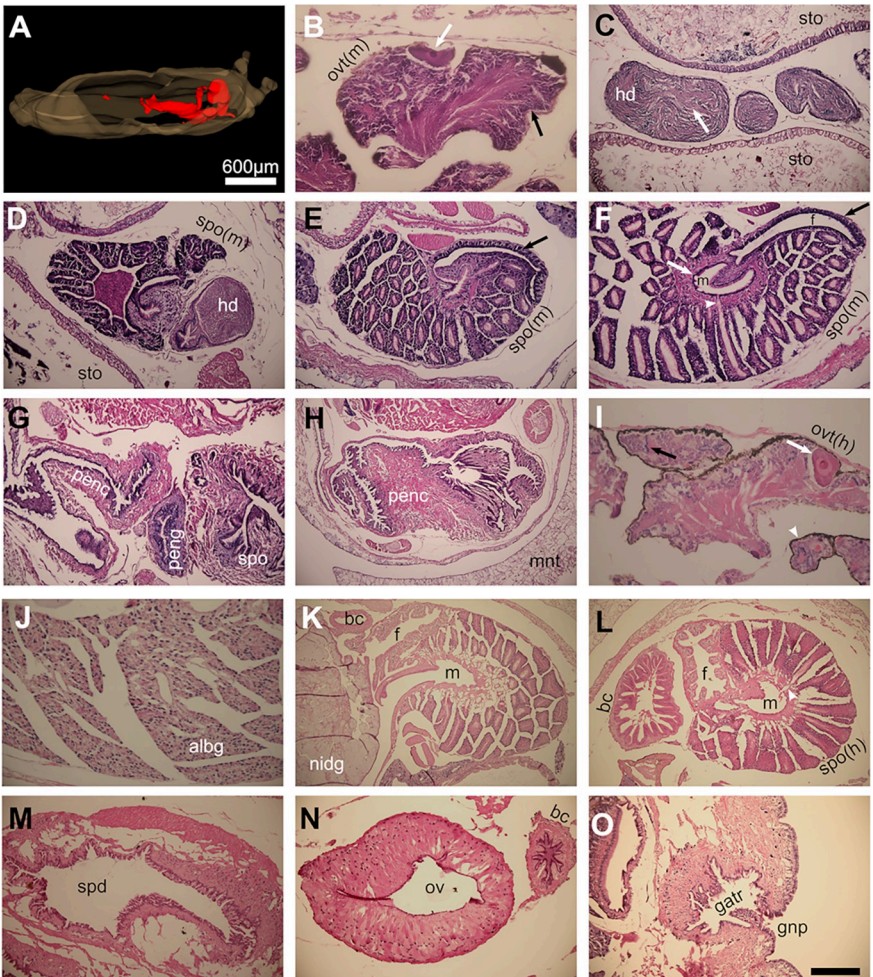

**Fig 16. Reproductive system of *A. valentianus*.** (A) 3D model showing the reproductive system in red. (B-I) Micrographs of a juvenile individual and (I-O) an older individual. (B) Ovotestis (predominantly spermatogenic). (C) Hermaphroditic duct. (D) Spermoviduct/prostate gland. (E) Spermoviduct/prostate gland. (F) Spermoviduct/prostate gland. (G) Spermoviduct and penis, (H) Penial complex. (I) Ovotestis (hermaphrodite). (J) Albumen gland. (K) Nidamental gland and spermoviduct/prostate gland. (L) Spermoviduct and bursa copulatrix. (M) Spermoviduct. (N) Oviduct and bursa copulatrix. (O) Genital atrium and gonopore. Values of scale bar: B and I (100 μm); C-G, J, and M-O (200 μm); H, K, and L (400 μm). Abbreviations in Table 1.

equivalent to the vas deferens, the apparent equivalent to the bursa copulatrix connects to the oviduct, and the penial gland appears hypotrophic (Fig 15B).

The reproductive tract of the older *A. valentianus* has hermaphroditic traits, thus resembling more that of *D. laeve* (Fig 15B'). The ovotestis still contains a large proportion of spermatogenic acini but it contains more oogenic cells with large nuclei, nucleoli and a more eosinophilic cytoplasm (Fig 16I). At this age, the albumen and the nidamental gland are also present, the former composed of cells with eosinophilic cytoplasm, in contrast to the unstained cytoplasm in *D. laeve* (Fig 16J and 16K). The hermaphrodite duct splits in two upon reaching the spermoviduct (Fig 15B'); the male duct of the latter displays the prostatic follicular appearance of the juvenile *A. valentianus* (Fig 16K and 16L), and the female duct is closer in appearance to the single duct spermoviduct of *D. laeve* albeit lacking the large basophilic cells (Fig 16K and 16L). Upon leaving the spermoviduct, the male duct connects to the sperm duct that leads to the penial complex and the female duct to the oviduct which merges with the penis

and continues into the genital atrium and genital pore (Figs 15B', 16M and 16N). The oviduct has a similar rosetta appearance of the equivalent structure in *D. laeve* except that instead of round, cells are more elongated and columnar (Fig 16N). As in *D. laeve*, the bursa copulatrix connects directly to the penial complex near its coalescence with the oviduct, and as in the juvenile *A. valentianus*, the topological equivalent to the penial gland appears vestigial and thin apparently corresponding to the flagellum (Fig 15B'). The merged oviduct and penial complex lead to the genital atrium and gonopore (Fig 16O).

## Tail regeneration in *Deroceras laeve*

To initiate studies of regeneration on *D. laeve*, we chose the tail which is easily accessible and can be of great use to elucidate the cellular and molecular mechanisms of its restoration. In a first series of studies, we amputated the tail in a way that only the body wall was affected including the epidermis and the subepidermal connective.

Within the first six hours to 2 days, cells of mesenchymal appearance are detected close to the exposed wound surface and a stratified layer of cells forms on its outer aspect with some cell delamination to the outside but no epidermis was observed (Fig 17A). This cell aggregate condenses by day 7, and at 14 days it becomes a clearly distinguishable structure akin to the blastema of other species and an epidermis now covers the amputation face (Fig 17B). The blastema at 7 days contains fibroblastoid cells, round cells resembling hemocytes, and muscle cells located deep in the lesion region (Fig 17B). Labeling with the marker of DNA synthesis F-ara-EdU revealed staining on the dorsal epidermis in non-amputated controls (Fig 17C) and

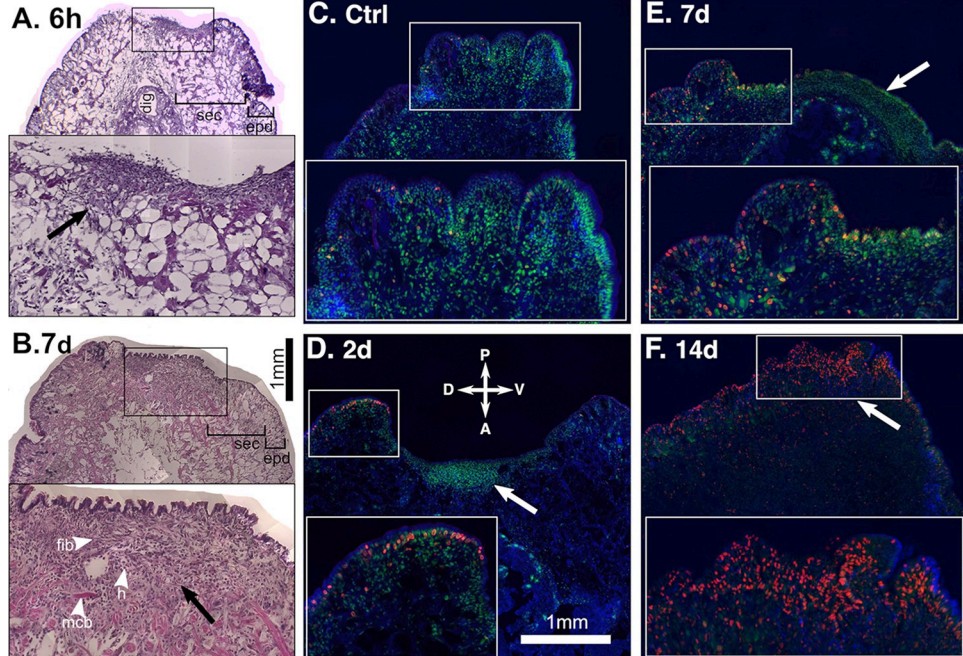

**Fig 17. Regeneration of the tail in *D. laeve*.** Sagittal sections after amputation: (A and B) stained with hematoxylin and eosin and (C-F) labeled with F-ara-EdU (red) counerstained with Hoechst 33342 (blue) and Sitox (green). Non-amputated control (Ctrl), 6 hours (6h) and days post-amputation (2d, 7d, and 14d) are shown in each panel. Rectangles in each panel indicate the magnified region shown below in each inset. Arrows indicate the location of the blastema. White arrowheads in B indicate fibroblastoid cells (f), muscle cells (m), and round cells resembling hemocytes (h). Dorsal (D), ventral (V), posterior (P), and anterior (A) orientations are indicated in D. Scale bar in B also applies to A and bar in D applies to C-F. Abbreviations in Table 1.

in the epidermis flanking the lesion on its dorsal side at 2 and 7 days post-amputation (Fig 17D and 17E). Deeper labeling at the amputation site was observed only after 14 days (Fig 17F). This suggests that the wound site is covered by an epidermis from the edges of the amputated face involving proliferation and that a blastema-like structure forms at the core of the regenerating tail in an early phase with no cell proliferation followed by a stage with proliferation. Regeneration is complete six to eight weeks after amputation.

## Discussion

As a first step to develop a study model of land gastropods, we established a colony of the stylommatophorans *D. leave* and *A. valentianus*. These species are easy to keep in the laboratory and at a very low cost, the former being more suitable as a study model owing to its smaller size, shorter time to reach reproductive age, higher hatching rate, and larger survival. This species has a high anatomical and physiological complexity compared to acoelomate invertebrates in which regeneration has thus far been studied and it possesses unique anatomical features. It has an open circulatory system with specialized cells, the hemocytes, it has a coelom, it has a high degree of cephalization of the nervous system, it has retractile tentacles, and has a high regenerative capacity. Moreover, the widespread global distribution of this species makes it amenable for study in many world regions.

As an essential element for the development of a gastropod model to study regeneration and other biological aspects, we built a web-based anatomical and histological resource of *D. leave* and *A. valentianus*, the SlugAtlas (https://slugatlas.lavis.unam.mx). The atlas provides histological images in three planes of section, 3D models of the major organs, and of the circumesophageal nerve ring of *D. laeve*. In this work we also provide a general histological description and comparative analysis of these species addressing their differences and little known or novel aspects.

A finding highly relevant from the perspective of stem cell biology is the degrowth and regrowth, observed in *D. laeve*, capabilities that have been observed previously in planarias which have a high regenerative potential [4–6, 39]. Decrease of 20% to 50% in dry mass, has also been observed in the freshwater pulmonate snail *Helisoma trivolvis* upon starvation [40], but no regrowth has been assessed in this species. Although further studies are required to understand the cellular and molecular basis of these phenomena in *D. laeve*, our results suggest that all tissues of the animal after degrowth can support cell proliferation and differentiation allowing the animal to restore its normal size upon feeding. This likely represents an adaptive advantage in coping with seasonal and prolonged scarcity of food in the wild.

Our histological survey of the suprapedal gland allowed a detailed description of its different elements. It also revealed the complex secretory cell network that surrounds it and what appear to be trans-cellular secretory streams that converge on the ventral side of the gland central duct and drain into it as observed in *Arion rufus* and *A. vulgaris* [32]. Moreover, ciliated cells were also found in the lumen epithelium as in *Helix* in which specific staining suggest the gland secretes neutral mucus [30].

Moreover, we confirmed the structure of the retractile posterior tentacles which are composed of a dermo-muscular outer tubular sheath and an inner muscular retractor tube with the sensory organs at their joint as in *Limax* and *Helix* [8, 30, 33]. Although the sensory structures are known to include photo-detection and olfactory elements, they are likely to carry tactile information as well. We also observed the previously described collar cells and we found them to be closely linked to the muscle cells that form the inner sheath via thin cell projections. The anterior tentacles, which are also retractile, revealed a similar arrangement with an outer dermo-muscular sheath but with the major difference that the retractor muscle lacks a tubular

structure and it is instead a muscular band that runs parallel to the tentacular nerve. Additionally, the adjacent Semper's organ was observed linked to a ciliated epidermal sensory plate with a nerve that joins the anterior tentacular nerve on its course to the cephalic ganglia. This contrasts with a previous description in *Limacus flavus* of a similar innervation of the anterior tentacle and Semper's organ by branches of the same nerve and an additional nerve innervating the tentacle [34]. The two main components of Semper's organ may be connected in such a way that sensory inputs from its ciliated plate may trigger exocrine or endocrine secretion via a local motor response, although central coordination by the cerebral ganglia may occur as well. This structure appears to be homologous to euthyneuran cephalic sensory organs (CSO) such as the lip organ or the anterior cephalic shield of opistobranchs as they are also innervated by a branch of the anterior tentacular nerve [41–43]. The dual sensory-glandular organization of Semper's organ, however, appears be a synapomorphy of the Superorder Eupulmonata as it is present in species of the Orders Stylommatophora and Systellomatophora [34, 44]. Moreover, the similarity of the collar cells of the tentacles to some cells of the secretory component of Semper's organ, supports the previously proposed notion that the former are also neurosecretory in nature [33, 34].

The integument and subepidermal tissue traversed by muscle cell bundles was found to be similar to that of *Aplysia* and *Helix*, and the proteinaceous appearance of the cytoplasm of Leydig cells similar to the equivalent cells in the connective tissue of *Lobatus gigas* [30].

Regarding the nervous system, in this work we provide a clear structure of the nerve ring based on a whole series of histological sections delineating clearly the cellular cortex of the ganglia, their neuropil, and connectives. This may serve as basis for a more thorough analysis of their connectivity and their peripheral targets.

Analysis of the buccal organ revealed its main components. We also confirmed the connection between the jaw and the radula by the cuticle that lines the oral cavity, the foregut cuticle as in *Ariolimax californicus* and *Cerithium sp.* [30, 35] forming a single chitinous structure resistant to SDS and proteinase K treatment.

Another finding in both species was the presence in the stomach of extracellular vesicles that shed into its lumen and are likely to contain digestive enzymes. The function of these structures that can be classified as microvesicles or ectosomes [45], however, remains to be elucidated.

Concretions of two types were also observed in our study. One type, located mostly in the subepidermal connective, may contain calcium carbonate and calcium phosphate, although other compounds may be present as well, with the notable finding of a high proportion of manganese. Calcium carbonate concretions have also been observed in cultured mantle cells and *in vivo* in hemocytes in the pearl oyster *Pinctada fucata*, in the mantle of the land slug *Arion rufus* and in tissues of the freshwater mussels *Anodonta anatina*, *Anodonta cygnea*, and *Unio pictorum* which have been implicated in the formation of the shell or its vestigial remnants [29, 46–48]. The other type of concretions was found in the kidney and are similar to those found in the snail *Helix* [30]. These concretions are heat labile and are linked to the nitrogenated excretory functions as they appear to be composed of uric acid as found for the pulmonate slugs *Planorbarius corneus* and *Elona quimperiana* [49, 50]. We also observed that these concretions reach the kidney lumen upon bursting of renocytes which puts them in their excretory pathway.

As for the overall complexity of the kidney, it was higher in *A. valentianus* than in *D. laeve*. The lamellar structure of the former was larger and more branched, on the whole occupying a relatively larger volume.

The structure of the heart in both species was similar to that of *Helix* in having a thin atrium traversed by muscle trabeculae and a more muscular ventricle [30]. Additionally, the heart was

found to undergo dramatic changes as the ventricle and the atrio-ventricular valve increase in thickness and complexity in older animals of both species, probably in parallel to the more demanding needs of the circulatory system in larger animals. Moreover, the flow of the hemolymph in the atrium is likely to be subdivided or irregular as many bundles of muscle trabeculae traverse it.

The lung was found to be clearly separate from the mantle cavity and its lining found to contain mounds and folds with tufts of microvilli. A close interaction of the respiratory system with the kidney was also observed in both species, further supported by the finding that the ureter drains into the airway on its way to the outside of the animal. Moreover, our 3D model allows the visualization of the close interaction of the heart, kidney, and lung in the dorsal aspect of the mantle complex.

Regarding the reproductive system, our study revealed that its topologically equivalent ductal components diverge in morphology and in function in the two species. Individuals of *D. laeve* were found to possess hermaphroditic features. In contrast, juveniles of *A. valentianus* had mostly male features as observed in *Cerithium sp*. while older individuals had hermaphroditic traits as in *D. laeve* [this work] and in *Helix* [30]. Such sex changes along life have been observed among Agriolimacidae [38]. This transition from male to hermaphrodite is known as protandry. Overall, our study provides histological insight into each ductal component of this system and a morphological correlate of protandry which can be of use to study its endocrine regulation.

The well known regenerative ability of gastropods was also put to a test by amputating the tail in adult *D. leave* animals. We observed that the amputation face is covered by a transition epidermis as observed in vertebrates [51–53] and in invertebrates [54]. Concomitantly, a cell condensation begins to form within the first few hours at the core of the tail that resembles the blastema of vertebrates and invertebrates [54–57]. Although similar in location and appearance, the time of formation of this structure differs from vertebrates and planaria. In amputated lizard tails it forms after 10 days [55], in limbs of lizard and salamanders it forms between 14 and 28 days [56], and in planaria it forms 2–3 days post-amputation [57]. Hence, regeneration initiates with similar steps as in vertebrates [51]. Namely, a hemostatic process closes the wound, re-epithelialization begins from the edges of the exposed tissues forming a wound epithelium, and a blastema-like structure is formed at the lesion site. A blastema containing hemocytes has also been described to form upon amputation of the cephalic tentacle of the caenogastropod *Pomacea canaliculata* [23, 24] and the arms of *Octopus* [58–61]. Further studies are now warranted to determine the nature, origin, and fate of the cells involved to assess conserved or divergent elements with other animal groups at the cellular and molecular level.

## Conclusions

We present SlugAtlas (https://slugatlas.lavis.unam.mx), a tool with high potential for terrestrial gastropod species that can be used to study diverse biological aspects such as stem cell biology, regeneration, and allometry. It is a resource that will support research groups interested in the study of these hitherto understudied subjects and is publicly available for research and educational purposes. In fact, an atlas such as the one we present herein can be an excellent support for inquiry-based science education as recently proposed for the undergraduate level using land gastropods [62]. The web-based platform of the atlas allows worldwide accessibility and it is hosted at the National Laboratory for Advanced Scientific Visualization (Laboratorio Nacional de Visualización Científica Avanzada, LAVIS-CONAHCYT) which ensures its long-term maintenance. Moreover, SlugAtlas is designed to accommodate more histological information and other contents, such as genomic data, for these and other species as we and other

interested research groups make them available. As proof of the usefulness of this resource, we performed a comparative histological analysis of *D. laeve* and *A. valentianus* and observed previously little known or unknown aspects or structures. We delved into unique anatomical features in these Agriolimacidae and Limacidae members, respectively, which will prompt additional studies with other members of the Superorder Eupulmonata including species of the sister Orders Stylommatophora and Systellommatophora. Of particular evolutionary relevance in these taxa is the structure of their retractile tentacles, the Semper's organ, and a potential equivalent of the suprapedal gland in Systellommatophora. This will address the homology and possible evolutionary divergence of these structures in connection to their limacization and terrestrial specialization. Furthermore, our work provides a solid basis for the study of regeneration with *D. laeve* and *A. valentianus* to fill the gap in the knowledge in stem cell biology, animal regeneration, and pluripotency of gastropods.

## Supporting information

**S1 Fig. Digital segmentation and blurring of slug organs.** (A) Example of segmentation. (B) Example of 3D blurring of the digestive tract. (C) Comparison between 3D models of the digestive tract without blurring and with blurring.
(TIF)

**S2 Fig. Concretions observed in *D laeve* and *A. valentianus*.** (A-G) Concretions observed in *D. laeve* and (H and H') in *A. valentianus*. (A and B) Bright-field micrographs of semithin sections of the body wall and (A' and B') the same fields with DIC optics showing refringent concretions (arrows). (C) Concretions extracted from the body wall that were heat-treated (concretions from different micrographs were combined in a single image and are shown at the same scale). (D) Transmission electron micrograph of concretions showing concentric rings in cross-section. (E and E') Histological section of the kidney showing concretions in the renal lumen in bright-field and DIC optics of the same field, respectively (arrows). (F) Scanning electron micrograph of a concretion extracted from the body wall. (G) Concretion extracted from the kidney. (H and H') Concretions observed in the subepidermal connective in bright-field and DIC optics of the same field in *A. valentianus* (arrows).
(TIF)

**S3 Fig. Energy dispersed X-ray spectroscopy (EDS) analysis of concretions of *D. laeve*.** (A) Concretions obtained from the body wall. (B) Concretions obtained from the kidney. Top micrographs on both, scanning electron microscopy images; bottom on both, EDS plot.
(TIF)

**S1 Table. Sequences used in the multiple alignment for the identification of *D. laeve* and *A. valentianus*.**
(DOCX)

**S2 Table. Energy dispersed X-ray spectroscopy (EDS) analysis of concretions in *D. laeve*.**
(DOCX)

**S1 Video. The various functionalities of the SlugAtlas to study the histological collections and 3D models of *D. laeve* are shown.**
(MP4)

**S2 Video. The various functionalities of the SlugAtlas to study the histological collections and 3D model of *A. valentianus* are shown.**
(MP4)

## Acknowledgments

Carlos Lozano Flores (Account number 43038 CONACYT) carried out the present work as a requirement of the Biochemistry PhD Program at UNAM (Programa de Doctorado en el Posgrado en Ciencias Bioquímicas, UNAM). For technical support we thank Luis A. Aguilar Bautista (Laboratorio Nacional de Visualización Científica Avanzada, LAVIS-CONAHCYT), Manuel Aguilar Franco (Laboratorio Nacional de Caracterización de Materiales, LaNCaM-CONAHCYT), Nidia Hernández Ríos, Erika de los Ríos Arellano, María Eugenia Rosas Alatorre, Omar González Hernández, Laura González Dávalos, and Mike Jeziorski. For support on the maintenace of the colony we thank Mayela Fosado Mendoza, Emilio Ortiz Ávila, Fernanda Rendón Montaudon, Mariana Rodríguez-Cabo Doria, Ilse Macedo Ramírez, Pamela Peña Medellín, and Alvaro Pablo Ricardo. We thank Jerónimo R. Miranda-Rodríguez for critical reading of the manuscript and Eduardo Varela for natural photography. Jorge Larriva-Sahd passed away before the submission of the final version of this manuscript. Alfredo Varela-Echavarría accepts responsibility for the integrity and validity of the data collected and analyzed.

## Author Contributions

**Conceptualization:** Carlos Lozano-Flores, Alfredo Varela-Echavarría.

**Data curation:** Carlos Lozano-Flores, Jessica Trujillo-Barrientos, Diego A. Brito-Domínguez, Rocío Cortés-Encarnación, Lizbeth D. Medina-Durazno, Alejandro de León-Cuevas, Alejandro Ávalos-Fernández, Wilbert Gutiérrez-Sarmiento, Fernando Javier Soto-Barragán, Gabriel Herrera-Oropeza, Ramón Martínez-Olvera, David Martínez-Acevedo, Luis C. Cruz-Blake, Vanessa Rangel-García, Alfredo Varela-Echavarría.

**Formal analysis:** Carlos Lozano-Flores, Fernando Javier Soto-Barragán, Gabriel Herrera-Oropeza, Alfredo Varela-Echavarría.

**Funding acquisition:** Jorge Larriva-Sahd, Remy Ávila, Alfredo Varela-Echavarría.

**Investigation:** Carlos Lozano-Flores, Jessica Trujillo-Barrientos, Diego A. Brito-Domínguez, Elisa Téllez-Chávez, Rocío Cortés-Encarnación, Lizbeth D. Medina-Durazno, Sergio Cornelio-Martínez, Wilbert Gutiérrez-Sarmiento, Aldo Torres-Barrera, Fernando Javier Soto-Barragán, Gabriel Herrera-Oropeza, Gema Martínez-Cabrera, Jorge Larriva-Sahd, Reinher Pimentel-Domínguez, Remy Ávila, Alfredo Varela-Echavarría.

**Methodology:** Carlos Lozano-Flores, Elisa Téllez-Chávez, Gema Martínez-Cabrera, Reinher Pimentel-Domínguez, Alfredo Varela-Echavarría.

**Project administration:** Alfredo Varela-Echavarría.

**Resources:** Alejandro de León-Cuevas, Alejandro Ávalos-Fernández, Alfredo Varela-Echavarría.

**Software:** Alejandro de León-Cuevas, Alejandro Ávalos-Fernández.

**Supervision:** Carlos Lozano-Flores, Alfredo Varela-Echavarría.

**Validation:** Carlos Lozano-Flores, Alfredo Varela-Echavarría.

**Visualization:** Carlos Lozano-Flores, Alfredo Varela-Echavarría.

**Writing – original draft:** Carlos Lozano-Flores, Alfredo Varela-Echavarría.

**Writing – review & editing:** Carlos Lozano-Flores, Reinher Pimentel-Domínguez, Remy Ávila, Alfredo Varela-Echavarría.

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
