## [Decision Letter · Decision Letter 0]

5 Aug 2024

PONE-D-24-25268SlugAtlas, a histological and 3D online resource of the land slugs Deroceras laeve and Ambigolimax valentianusPLOS ONE

Dear Dr. Varela-Echavarría,

Thank you for submitting your manuscript to PLOS ONE. After careful consideration, we feel that it has merit but does not fully meet PLOS ONE’s publication criteria as it currently stands. Therefore, we invite you to submit a revised version of the manuscript that addresses the points raised during the review process.

Your manuscript has been evaluated by two reviewers, who have expressed overall positive opinions about your research. However, they have raised several important points that require your attention. We ask that you address each of these points comprehensively in your revision. To strengthen your paper, I recommend considering a broader context for your work. Specifically, you might consider including comparisons with other gastropod molluscs, particularly the well-studied Aplysia. This could provide a richer framework for your findings and enhance their significance within the field.

In preparing your revision, please ensure that your responses to the reviewers' comments are reflected in the manuscript itself. This approach will allow readers to benefit from the additional information, and ensure that the manuscript is improved in light of the reviewers' comments and questions.

We look forward to receiving your revised manuscript.

Kind regards,

Jian Jing, Ph.D.

Academic Editor

PLOS ONE

3. We note that Figure(s) 1a,b, 3 to 16, S2, and S3 in your submission contain copyrighted images. All PLOS content is published under the Creative Commons Attribution License (CC BY 4.0), which means that the manuscript, images, and Supporting Information files will be freely available online, and any third party is permitted to access, download, copy, distribute, and use these materials in any way, even commercially, with proper attribution. For more information, see our copyright guidelines: http://journals.plos.org/plosone/s/licenses-and-copyright.

a. You may seek permission from the original copyright holder of Figure(s) 1a,b, 3 to 16, S2, and S3 to publish the content specifically under the CC BY 4.0 license. 

Reviewers' comments:

Reviewer's Responses to Questions

**Comments to the Author**

1. Is the manuscript technically sound, and do the data support the conclusions?

Reviewer #1: Partly

Reviewer #2: Yes

2. Has the statistical analysis been performed appropriately and rigorously? 

Reviewer #1: Yes

Reviewer #2: N/A

3. Have the authors made all data underlying the findings in their manuscript fully available?

Reviewer #1: Yes

Reviewer #2: Yes

4. Is the manuscript presented in an intelligible fashion and written in standard English?

Reviewer #1: Yes

Reviewer #2: Yes

5. Review Comments to the Author

Reviewer #1: In this work the authors performed the first characterization of whole adult individuals of both D. laeve and A. valentianus in three planes of section and described aspects of growth and maintenance of them. The digitalized imaging data were compiled into an online server-based anatomical atlas.

While collecting valuable anatomical data, the critical claims in justifying the significance of the dataset as paving the road towards new model organism in studying regeneration should be further justified. In the work, the authors performed dietary changes that induced degrowth and regrowth of the animal, and in sea slugs it has been recently found that fully body could have regenerated with great capacity. Thus, a body amputation and regeneration inspection (with potentially EdU staining for cell growth) of D. laeve should be conducted to fully justify that land slug is a group of organisms capable of self-regeneration. The regeneration related genes should be aligned and inspected with other self-regenerating animals to leverage bioinformatics for further getting how confident we may believe land slugs as a suitable model organisms for regeneration.

Currently the SlugAtlas website contains only histological data, and the phylogenetic studies performed in the work should be conveniently visualized, as genome and evolutionary information is highly desirable for molecular biology studies.

Potentially beyond the scope of the capacity of this revision, the capability and protocol for producing modified clones of organism with RNAi/CRSIPR is also crucial for any molecular perturbation of the animal. A de novo assembly of transcriptome and the profiling of gene expression dynamics during degrowth and regrowth will also provide more insights translatable into comparative development biology and regeneration studies in the nowadays research context.

Reviewer #2: This is a detailed and thorough report of histological sections through the body of two slugs in the Order Stylommatophora. The quality of the data is high, and the online Slug Atlas is a potentially valuable resource. It is simple to access this digital atlas and the organs are fairly clear. My major reservation is the limited accessibility. For example, there is no discussion of the relationship between these two pulmonate slugs and other heterobranchs.

According to Wikipedia, “The two strong synapomorphies of Stylommatophora are a long pedal gland placed beneath a membrane and two pairs of retractile tentacles.” We should be told this at the outset of the paper.

A number of terms merit explanation early on in the MS. Degrowth, which is a simple concept, should be clearly explained.

Some anatomical structures deserve introduction early in Results or in the Introduction. For example: suprapedal gland and Semper’s organ (is this a neurosecretory gland?).

Line 343 “sexual dimorphism found only in A. valentianus.” This seems to be a confused description as both species are hermaphrodites. (On line 895, this is apparently clarified, “juvenile A. valentianus appeared to have mostly male features and the older individuals had hermaphroditic traits,” but this should be explained earlier.). Are the juvenile A. valentianus sexually active, or is this simply a developmental stage where they are sexually immature?

Line 353. “sensory neurons” in Figure 6. This term is only used here, and there is no indication in the micrographs of Fig 6 as to the location of sensory neurons. This also raises the question of how neurons were identified.

Below are a number of minor wording comments.

Line 42. “specially slugs” should perhaps be “especially slugs” or this might be clearer as “This is especially true of slugs, which lack ..”

Line 53. “and associated to the air exchange pore” might read better as “adjacent to the air exchange pore”

Line 72. “Atlantic Islands” are defined differently depending on the reference, so it would be helpful to list them specifically. Is this Canaries, Madeira, Azores and Cape Verde Islands

Line 102 “a saturated solution of menthol in ethanol.” I could not understand this, as these two alcohols are totally miscible in water.

6. PLOS authors have the option to publish the peer review history of their article (what does this mean?). If published, this will include your full peer review and any attached files.

Reviewer #1: No

Reviewer #2: No

---

## [Author Response · Author response to Decision Letter 0]

19 Aug 2024

Editor’s comments: 

COMMENT: To strengthen your paper, I recommend considering a broader context for your work. Specifically, you might consider including comparisons with other gastropod molluscs, particularly the well-studied Aplysia. This could provide a richer framework for your findings and enhance their significance within the field. 

RESPONSE: Additional information has been included in the Introduction referencing examples of other gastropod models used for different research purposes. Moreover, throughout Discussion, results of this study are now contrasted with other sea and land gastropods including Aplysia and with some bivalves as well. Also, the morphological aspects of the regeneration of the tail, are compared with those of vertebrates and other invertebrates including planarias, the snail Pomacea canaliculata, the cephalopod Octopus, and Aplysia. 

COMMENT: RESPONSE: An updated version of the financial disclosure with funding information has been included at the end of the cover letter. 

COMMENT: Guidelines for resubmitting your figure files are available below the reviewer comments at the end of this letter.

RESPONSE: All main figures have been edited to comply with the journal ´s guidelines, uploaded to the Preflight Analysis and Conversion Engine (PACE) digital diagnostic tool, and resubmitted. 

COMMENT: If applicable, we recommend that you deposit your laboratory protocols in protocols.io to enhance the reproducibility of your results.

RESPONSE: We plan to submit at a later time a separate protocols article with the methods that are specific for these model animals.

Reviewers' comments:

Reviewer #1

COMMENT: In this work the authors performed the first characterization of whole adult individuals of both D. laeve and A. valentianus in three planes of section and described aspects of growth and maintenance of them. The digitalized imaging data were compiled into an online server-based anatomical atlas.

While collecting valuable anatomical data, the critical claims in justifying the significance of the dataset as paving the road towards new model organism in studying regeneration should be further justified. In the work, the authors performed dietary changes that induced degrowth and regrowth of the animal, and in sea slugs it has been recently found that fully body could have regenerated with great capacity. Thus, a body amputation and regeneration inspection (with potentially EdU staining for cell growth) of D. laeve should be conducted to fully justify that land slug is a group of organisms capable of self-regeneration. The regeneration related genes should be aligned and inspected with other self-regenerating animals to leverage bioinformatics for further getting how confident we may believe land slugs as a suitable model organisms for regeneration.

RESPONSE: In agreement with the reviewer’s suggestion of including an exploration of regeneration in this model system, we have added a description of the morphological aspects of regeneration of the tail of Deroceras laeve upon amputation as well as an inspection of cell proliferation using the marker of DNA synthesys F-ara- EdU (Figure 17). This also required the inclusion of new sections or paragraphs in Materials and Methods (Amputation and F-ara-EdU labeling, line 128), Results (Tail regeneration in Deroceras laeve, line 873), Discussion (line 1034), and Conclusions (line 1075).

We also coincide in that performing molecular analysis will allow comparative studies with other animals capable of regeneration. One current difficulty is that a transcriptome is not yet available for D. laeve. For this reason we have initiated analysis of the molecular processes involved in the regeneration of the tail entailing a de novo assembly of a transcriptome which will continue beyond the scope of the current work. 

COMMENT: Currently the SlugAtlas website contains only histological data, and the phylogenetic studies performed in the work should be conveniently visualized, as genome and evolutionary information is highly desirable for molecular biology studies.

RESPONSE: We included in the SlugAtlas webpage the results of the phylogenetic analysis described in the manuscript along with links to pages with information on the stylommatophoran species employed in such analysis. Additionally, SlugAtlas will incorporate references to transcriptomic and genomic data as we make it available and this is stated now in Conclusions. 

COMMENT: Potentially beyond the scope of the capacity of this revision, the capability and protocol for producing modified clones of organism with RNAi/CRSIPR is also crucial for any molecular perturbation of the animal. A de novo assembly of transcriptome and the profiling of gene expression dynamics during degrowth and regrowth will also provide more insights translatable into comparative development biology and regeneration studies in the nowadays research context. 

RESPONSE: In agreement with the reviewer, although beyond the scope of this manuscript, we are currently performing a study on the cellular and gene control of degrowth and regrowth employing a transcriptomic approach. Efforts are also underway to develop gene perturbation methods that will continue in subsequent studies. 

Reviewer #2:

COMMENT: This is a detailed and thorough report of histological sections through the body of two slugs in the Order Stylommatophora. The quality of the data is high, and the online Slug Atlas is a potentially valuable resource. It is simple to access this digital atlas and the organs are fairly clear. My major reservation is the limited accessibility. For example, there is no discussion of the relationship between these two pulmonate slugs and other heterobranchs.

RESPONSE: Following the reviewer's observation, we have included in the first paragraph of the Introduction the following two sentences: “The Order Stylommatophora is the most abundant group of land slugs and snails. It belongs to the Subclass Heterobranchia also encompassing marine slugs and snails” (line 42).

Moreover, throughout Discussion, results with D. laeve and A. valentianus are contrasted with those of other heterobranchs, caenogastropods, bivalves, and cephalopods. 

COMMENT: According to Wikipedia, “The two strong synapomorphies of Stylommatophora are a long pedal gland placed beneath a membrane and two pairs of retractile tentacles.” We should be told this at the outset of the paper.

RESPONSE: We now state clearly that retractile tentacles and a suprapedal gland isolated from the visceral cavity are stylommatophoran synapomorphies (line 50). 

COMMENT: A number of terms merit explanation early on in the MS. Degrowth, which is a simple concept, should be clearly explained.

RESPONSE: Following the reviewer's suggestion, the following terms have been defined in the text: 

limacization (line 57)

degrowth /regrowth (lines 65-67) cleptoplasty (line 73)

allometry (lines 105-106)

F-actin staining with phalloidin (line 467) diaulic (line 841)

protandry (line 1030) 

COMMENT: Some anatomical structures deserve introduction early in Results or in the Introduction. For example: suprapedal gland and Semper’s organ (is this a neurosecretory gland?).

RESPONSE: Following the reviewer's suggestion, the following structures have been described the first time they are mentioned in Introduction or Results: 

rhinophore (line 53)

ommatophore (line 55)

pneumostome (lines 60, 439)

radula (line 273)

circumesophageal nerve ring (line 381) integument (line 407) 

subepidermal connective (line 410) mantle (line 434)

free mantle (line 436)

concretions (line 446) 

ovotestis (line 797)

Semper’s organ. Indeed, as stated in the manuscript (line 554), it has been proposed that this organ is neurosecretory. We speculate that its two main components may be connected in such a way that sensory inputs from its external ciliated plate may trigger exocrine or endocrine secretion via a local motor response, although central coordination by the cerebral ganglia may occur as well. 

COMMENT: Line 343 “sexual dimorphism found only in A. valentianus.” This seems to be a confused description as both species are hermaphrodites. (On line 895, this is apparently clarified, “juvenile A. valentianus appeared to have mostly male features and the older individuals had hermaphroditic traits,” but this should be explained earlier.). Are the juvenile A. valentianus sexually active, or is this simply a developmental stage where they are sexually immature? 

RESPONSE: We have eliminated the use of the term “Sexual dimorphism” as it is indeed confusing and in Discussion we described in better detail the differences in the sexual organs of A. valentianus in different stages of life also defining the term "protandry" (lines 1024-1032). Juveniles of Agriolimacidae have been observed to be sexually mature functioning predominantly as males producing sperm cells while older animals produce both male and female gametes (Wiktor, 2000). 

COMMENT: Line 353. “sensory neurons” in Figure 6. This term is only used here, and there is no indication in the micrographs of Fig 6 as to the location of sensory neurons. This also raises the question of how neurons were identified.

RESPONSE: Reference to sensory neurons was eliminated as no specific staining method for this type of cells was used in this study. 

Wording comments. 

COMMENT: Line 42. “specially slugs” should perhaps be “especially slugs” or this might be clearer as “This is especially true of slugs, which lack ..”

RESPONSE: The suggested change has been made (line 45). 

COMMENT: Line 53. “and associated to the air exchange pore” might read better as “adjacent to the air exchange pore”

RESPONSE: The suggested change has been made (line 60). 

COMMENT: Line 72. “Atlantic Islands” are defined differently depending on the reference, so it would be helpful to list them specifically. Is this Canaries, Madeira, Azores and Cape Verde Islands

RESPONSE: Since the oldest reports of this species are from the Iberian peninsula and the reports in diverse islands are more recent and likely to be secondarily introduced, we have eliminated reference to the latter (line 81). 

COMMENT: Line 102 “a saturated solution of menthol in ethanol.” I could not understand this, as these two alcohols are totally miscible in water.

RESPONSE: Menthol is a monoterpenoid which as a white crystalline solid has low solubility in water. For that reason we made a saturated solution adding an excess of menthol to absolute ethanol such that an insoluble precipitate remained after vigorous mixing. This allowed us to measure small amounts to dilute in water 1:400 as described in the manuscript. A more detailed description of the procedure is now included (lines 123-126).

---

## [Decision Letter · Decision Letter 1]

29 Sep 2024

PONE-D-24-25268R1SlugAtlas, a histological and 3D online resource of the land slugs Deroceras laeve and Ambigolimax valentianusPLOS ONE

Dear Dr. Varela-Echavarría,

Thank you for submitting your manuscript to PLOS ONE. After careful consideration, we feel that it has merit but does not fully meet PLOS ONE’s publication criteria as it currently stands. Therefore, we invite you to submit a revised version of the manuscript that addresses the points raised during the review process.

One reviewer raised mostly minor issues. Please make necessary changes.

We look forward to receiving your revised manuscript.

Kind regards,

Jian Jing, Ph.D.

Academic Editor

PLOS ONE

Journal Requirements:

Reviewers' comments:

Reviewer's Responses to Questions

**Comments to the Author**

1. If the authors have adequately addressed your comments raised in a previous round of review and you feel that this manuscript is now acceptable for publication, you may indicate that here to bypass the “Comments to the Author” section, enter your conflict of interest statement in the “Confidential to Editor” section, and submit your "Accept" recommendation.

Reviewer #1: All comments have been addressed

Reviewer #2: (No Response)

2. Is the manuscript technically sound, and do the data support the conclusions?

Reviewer #1: Yes

Reviewer #2: Yes

3. Has the statistical analysis been performed appropriately and rigorously? 

Reviewer #1: Yes

Reviewer #2: Yes

4. Have the authors made all data underlying the findings in their manuscript fully available?

Reviewer #1: Yes

Reviewer #2: Yes

5. Is the manuscript presented in an intelligible fashion and written in standard English?

Reviewer #1: Yes

Reviewer #2: Yes

6. Review Comments to the Author

Reviewer #1: The authors has made significant improvement in the revision by providing background in other gastropod models in the introduction and contrast with other sea and land gastropods including Aplysia and with some bivalves are included in discussion. Moreover, morphological aspects of regeneration of the tail of Deroceras laeve upon amputation as well as an inspection of cell proliferation using the marker of DNA synthesys F-ara- EdU are completing the data's significance in paving the road toward establishing a new model organism for regeneration.

Reviewer #2: This revised paper is far more accessible. The description of the order Stylommatophora is very helpful, and the explanations of the organ systems are clear. The tail regeneration section is a very nice addition.

I have a number of minor wording concerns.

A very minor point is that there seems to be excessive use of abbreviations. Perhaps some of these are primarily used in the atlas, but if an abbreviation is not used at least 2-3 times in the text, there is no benefit to using the abbreviation in the MS, as opposed to the anatomical atlas. One clear example is sec, an abbreviation for subepidermal connective; this is an unnecessary abbreviation, as it may only be used a single time. For example, “only the body wall was affected including the epidermis and the sec” (line 877) is confusing.

Line 89. “been a valuable tool … in recent works transcriptomic resources of its nervous system and for embryological studies have been developed.” These two areas – transcriptomic resources of its nervous system and “for embryological studies” – are not parallel. This sentence needs to be rewritten.

Line 104. “Hence, it is possible to envision that this resource will be of use to study in gastropods, among other subjects, anatomy, stem cell biology, regeneration, embryology, and the control of body proportions or allometry.” This sentence should be rewritten, as it does not quite make sense. Perhaps it could be worded “Hence, this resource will potentially be useful in various areas of research in gastropods, including anatomy, stem cell biology, regeneration, embryology, and allometry, the control of body proportions.”

Line 130. “clean cut of a razor blade, after which were transferred”. This sentence needs a subject before “were transferred,” such as “they.”

Line 555. The Semper’s organ is a dual sensory-glandular structure that contains a ganglion connected with a subepidermal sensory component with a glomerular organization akin to that of the tentacular digitiform ganglia, linked to a ciliated epithelial plate facing the ventral side of the animal at the mouth lobes just anterior to the prebuccal groove.” This should be split into two or three sentences, and the wording clarified.

Line 1030. “and transition from male to hermaphrodite is known as protandry” should be a separate sentence, e.g. “This transition from male to hermaphrodite is known as protandry.”

Line 1035. “We observed that the amputated face is covered by a transition epidermis” should probably be “amputation face.” This is a subtle difference, but as written it suggests that a face was surgically removed.

7. PLOS authors have the option to publish the peer review history of their article (what does this mean?). If published, this will include your full peer review and any attached files.

Reviewer #1: No

Reviewer #2: No

---

## [Author Response · Author response to Decision Letter 1]

3 Oct 2024

PONE-D-24-25268

SlugAtlas, a histological and 3D online resource of the land slugs Deroceras laeve and Ambigolimax valentianus

PLOS ONE

Editor’s comments: 

COMMENT: Please review your reference list to ensure that it is complete and correct. If you have cited papers that have been retracted, please include the rationale for doing so in the manuscript text, or remove these references and replace them with relevant current references. Any changes to the reference list should be mentioned in the rebuttal letter that accompanies your revised manuscript. If you need to cite a retracted article, indicate the article’s retracted status in the References list and also include a citation and full reference for the retraction notice.

RESPONSE: The reference list has been confirmed to be complete and correct.

Comments of Reviewer #2

COMMENT: A very minor point is that there seems to be excessive use of abbreviations. Perhaps some of these are primarily used in the atlas, but if an abbreviation is not used at least 2-3 times in the text, there is no benefit to using the abbreviation in the MS, as opposed to the anatomical atlas. One clear example is sec, an abbreviation for subepidermal connective; this is an unnecessary abbreviation, as it may only be used a single time. For example, “only the body wall was affected including the epidermis and the sec” (line 877) is confusing.

RESPONSE: We have reduced the use of abbreviations throughout the text and only use them to label structures in the figures. The abbreviated form of “sec” is provided on its first mention in the main text in reference to Figure 6 (line 410) and in its Legend (Line 430). The spelled out form, however, is used throughout the text (lines 414, 419, 430, 431, 443, 447, 877, and 992).

COMMENT: Line 89. “been a valuable tool … in recent works transcriptomic resources of its nervous system and for embryological studies have been developed.” These two areas – transcriptomic resources of its nervous system and “for embryological studies” – are not parallel. This sentence needs to be rewritten.

RESPONSE: The suggested change has been made (line 86).

COMMENT: Line 104. “Hence, it is possible to envision that this resource will be of use to study in gastropods, among other subjects, anatomy, stem cell biology, regeneration, embryology, and the control of body proportions or allometry.” This sentence should be rewritten, as it does not quite make sense. Perhaps it could be worded “Hence, this resource will potentially be useful in various areas of research in gastropods, including anatomy, stem cell biology, regeneration, embryology, and allometry, the control of body proportions.”

RESPONSE: The suggested change has been made (line 103).

COMMENT: Line 130. “clean cut of a razor blade, after which were transferred”. This sentence needs a subject before “were transferred,” such as “they.”

RESPONSE: The suggested change has been made (line 130).

COMMENT: Line 555. The Semper’s organ is a dual sensory-glandular structure that contains a ganglion connected with a subepidermal sensory component with a glomerular organization akin to that of the tentacular digitiform ganglia, linked to a ciliated epithelial plate facing the ventral side of the animal at the mouth lobes just anterior to the prebuccal groove.” This should be split into two or three sentences, and the wording clarified.

RESPONSE: The suggested change has been made (line 554).

COMMENT: Line 1030. “and transition from male to hermaphrodite is known as protandry” should be a separate sentence, e.g. “This transition from male to hermaphrodite is known as protandry.”

RESPONSE: The suggested change has been made (line 1028).

COMMENT: Line 1035. “We observed that the amputated face is covered by a transition epidermis” should probably be “amputation face.” This is a subtle difference, but as written it suggests that a face was surgically removed.

RESPONSE: The suggested change has been made (line 1035).

---

## [Editor Report · Decision Letter 2]

7 Oct 2024

SlugAtlas, a histological and 3D online resource of the land slugs Deroceras laeve and Ambigolimax valentianus

PONE-D-24-25268R2

Dear Dr. Varela-Echavarría,

We’re pleased to inform you that your manuscript has been judged scientifically suitable for publication and will be formally accepted for publication once it meets all outstanding technical requirements.

Kind regards,

Jian Jing, Ph.D.

Academic Editor

PLOS ONE
---

## [Editor Report · Acceptance letter]

10 Oct 2024

PONE-D-24-25268R2 

PLOS ONE

Dear Dr. Varela-Echavarría, 

I'm pleased to inform you that your manuscript has been deemed suitable for publication in PLOS ONE. Congratulations! Your manuscript is now being handed over to our production team.

Kind regards, 

on behalf of

Dr Jian Jing 

Academic Editor

PLOS ONE